# New learning while consolidating memory during sleep is actively blocked by a protein synthesis dependent process

**Roi Levy\*, David Levitan, Abraham J Susswein\***

The Leslie and Susan Gonda (Goldschmied) Multidisciplinary Brain Research Center, The Mina and Everard Goodman Faculty of Life Sciences, Bar Ilan University, Ramat Gan, Israel

**Abstract** Brief experiences while a memory is consolidated may capture the consolidation, perhaps producing a maladaptive memory, or may interrupt the consolidation. Since consolidation occurs during sleep, even fleeting experiences when animals are awakened may produce maladaptive long-term memory, or may interrupt consolidation. In a learning paradigm affecting *Aplysia* feeding, when animals were trained after being awakened from sleep, interactions between new experiences and consolidation were prevented by blocking long-term memory arising from the new experiences. Inhibiting protein synthesis eliminated the block and allowed even a brief, generally ineffective training to produce long-term memory. Memory formation depended on consolidative proteins already expressed before training. After effective training, long term memory required subsequent transcription and translation. Memory formation during the sleep phase was correlated with increased CREB1 transcription, but not CREB2 transcription. Increased C/EBP transcription was a correlate of both effective and ineffective training and of treatments not producing memory.

**\*For correspondence:** roiv.levy@gmail.com (RL); avy@biu.ac.il (AJS)

**Competing interests:** The authors declare that no competing interests exist.

## Introduction

New experiences that occur while a memory is being consolidated interact with the consolidation. The interactions can lead to problems in keeping apart related memories. This problem is partially solved by deferring some of the memory consolidation to sleep, a period in which new experiences are minimized. However, experiences when animals are wakened from sleep are likely to interact with the ongoing consolidation, thereby potentially causing maladaptive memories. We have found that during the sleep phase consolidation, new experiences produce a protein synthesis dependent process that blocks the formation of new memories, thereby preventing the formation of maladaptive memories.

Some of the potential problems that arise from successive experiences have been known for many years. An experience that occurs while a memory is being consolidated may interfere with the consolidation, and partially stops it. Indeed, the process of memory consolidation was discovered over 100 years ago by interference of memory formation as a result of a new learning trial (*Lechner et al., 1999*). Only much later was consolidation shown to depend on gene transcription and translation (*Montarolo et al., 1986*; *Sweatt, 2010*), and dependence on these molecular processes has become the defining feature of consolidation. A second interaction between an experience and an ongoing consolidation has been more recently discovered. An experience that is generally ineffective in producing long-term memory can become effective when it occurs during the consolidation of a memory that arises from another, effective experience. This phenomenon was originally described on a synaptic level, and was called synaptic tagging and capture, since training

**eLife digest** Throughout our waking lives we are exposed to a continuous stream of experiences. Some of these experiences trigger changes in the strength of connections between neurons in the brain and begin the process of forming memories. However, these initial memory traces are fragile and only a small number will become long-term memories with the potential to last a lifetime. For this transition to occur, the brain must stabilize the memory traces through a process called consolidation.

During consolidation, the brain produces new proteins that strengthen the fragile memory traces. However, if a new experience occurs while an existing memory trace is being consolidated, the new experience could disrupt or even hijack the consolidation process. To avoid this problem, the brain performs most consolidation while we are asleep. But what happens if we wake up while consolidation is taking place? How does the brain prevent events that occur just after awakening from disrupting consolidation?

Levy et al. have now answered this question using a seemingly unlikely subject, the sea slug *Aplysia*. Sea slugs are capable of basic forms of learning, and their simple nervous systems and large neurons make them convenient to study. Blocking the production of new proteins in sleeping sea slugs prevents the animals from forming long-term memories, confirming that, like us, they do consolidate memories during sleep. Levy et al. now show that exposing sea slugs to new stimuli immediately after they wake up does not trigger the formation of new memories. However, if the slugs were treated with a drug that blocks protein production just beforehand, the new stimuli could trigger memory formation.

These findings show that proteins blocking the formation of new memories prevent an experience upon waking from being effective in producing memory. Removing this block – by inhibiting protein production – allows experiences just after waking to be encoded in memory. This even applies to experiences that are too brief to trigger memory formation in fully awake sea slugs. The next step following on from this work is to identify these memory blocking proteins and to work out how they prevent new memories from forming. A future challenge is to find out is whether the same proteins could ultimately be used to block unwanted memories.

at synapses that have undergone short-term plasticity produces local tags (*Frey and Morris, 1997*; *Martin et al., 1997*). These tags can capture molecular products that are synthesized as a result of a consolidation process that arises from another training. Capture at the tags causes expression of long-term memory at synapses that would otherwise only display short-term memory. More recent studies have demonstrated behavioral tagging that is analogous to the synaptic tagging (*Ballarini et al., 2009*; *de Carvalho Myskiw et al., 2013*; *Moncada and Viola, 2007; Moncado et al., 2015*).

The problem of potentially non-adaptive interactions between memory consolidation and new experiences may be particularly acute when animals sleep. There is now abundant evidence that after completion of the early stages of consolidation which follow an experience, an additional stage of consolidation occurs while animals sleep (*Rasch and Born, 2013*; *Walker and Stickgold, 2006*), primarily during the night for diurnally active animals, or during the day for nocturnally active animals. Similar processes may govern consolidation that immediately follows training, and that during sleep. Memory consolidation and cortical plasticity during sleep are also accompanied by, or is dependent on protein synthesis (*Grønli et al., 2013*; *Rasch and Born, 2013*; *Seibt et al., 2012*; *Tudor et al., 2016*; *Walker, 2005*). There is also a general increase in cerebral protein synthesis during sleep phases associated with memory consolidation (*Ramm and Smith, 1990*). In addition, molecular events associated with memory consolidation during sleep are similar to those induced during consolidation following training (*e.g.*, *Luo et al., 2013*). Consolidation-like processes occur during sleep even in the absence of a previous training trial. Animals and humans sleep every day, even when no overt training occurred in the previous active phase. Proteins supporting consolidation will be synthesized during sleep, even when sleep does not follow overt training (*Ramm and Smith, 1990*).

As in consolidation that follows training, new experiences during the sleep-phase may cause non-adaptive interaction with the consolidation. A transient experience that would not cause long-term memory could be amplified by the ongoing consolidation, and cause an unwanted long-term memory. The amplification of the experience could then in turn increase the possibility that the new experience will interfere with consolidation of the earlier memory during sleep. One way of preventing interference is to prevent new experiences by losing consciousness. Indeed, the earliest papers (*Jenkins and Dallenbach, 1924*) showing that sleep improves memory posited that it does so passively, by preventing interference from new experiences. In recent years it has become clear that active processes also strengthen memories while animals sleep (*Rasch and Born, 2013*; *Walker and Stickgold, 2006*). However, the presence of active processes promoting consolidation during sleep strengthens the need to limit experiences, or the potential memories elicited by such experiences, while memories are consolidated. Animals and humans often wake from sleep, and experiences during such times should interfere with consolidation. However, paradoxically, training during the inactive phase, when animals are awakened from sleep, is often relatively ineffective in producing memory (*Lyons et al., 2005*; *Michel and Lyons, 2014*; *Page, 2015*; *Rawashdeh et al., 2007*).

We explored the possibility that training during the sleep phase initiates an active block of learning, as a means of preventing such inappropriate learning from being consolidated, or of interfering with consolidation. If such a block is present, one would predict that if the block is removed, experiences adjacent to sleep should become more effective in initiating long-term memory. Under this condition, experiences that do not cause long-term memory when animals are active should cause long-term memory.

To explore mechanisms by which new experiences during the period when animals generally sleep are prevented from interfering with consolidation, we utilized an associative learning paradigm in *Aplysia* in which animals learn that a tasty food cannot be swallowed (*Botzer et al., 1998*; *Katzoff et al., 2002*; *2006*, *Lyons et al., 2005*; *Michel et al., 2013*; *Susswein et al., 1986*). Training produces a gradual decrease in responsiveness to food, and animals stop responding after 10–25 min of training. Training during the active phase of the day produces short, intermediate, long-term (*Botzer et al., 1998*; *Michel et al., 2013*), and persistent (*Levitan et al., 2012*; *Schwarz et al., 1991*) memories, with the long-term memory dependent on protein synthesis adjacent to the training and persistent memory dependent on a somewhat later round of protein synthesis (*Levitan et al., 2010*). Training during the inactive, sleep phase of the day is ineffective in producing long-term memory (*Lyons et al., 2005*). We now report that training during the sleep phase can be made effective by treatment with a short-acting inhibitor of protein synthesis. This treatment permits even experiences that are too brief to cause long-term memory to become effective. Thus, protein synthesis following training has opposite effects on memory resulting from training during the active or the sleep phase, blocking the former while enabling the latter. Our findings indicate that training during the sleep phase initiates an active protein-synthesis dependent process that blocks the formation of new long-term memories. In addition, when the active block is removed, long-term memory formation is no longer dependent on protein synthesis at the time of the training. However, long-term memory formation is dependent on transcription and translation a number of hours after training.

We examined a number of molecular correlates of long-term memory formation during the sleep phase. Changes were examined in the transcription of *Aplysia* CREB1, CREB2 and C/EBP, molecules that are associated with memory formation (*Alberini, 2009*). We found no changes in CREB2 expression. Treatments that did and that did not cause memory formation led to increased C/EBP expression, indicating that C/EBP expression is not sufficient for memory formation. By contrast, only a treatment causing long-term memory caused a significant increase in the expression of CREB1, indicating that this increase is a correlate of memory formation.

## Results

A previous report (*Lyons et al., 2005*) showed that training *Aplysia* during their sleep phase is ineffective in producing long-term memory. Our aim was to test the hypothesis that this training is ineffective, because it is adaptive to prevent new learning during an ongoing process of memory consolidation that occurs during sleep. If the training during sleep phase were effective two problems would arise: (1) the molecular events underlying consolidation could enhance the efficacy of the

training at this time, leading to dysfunctional long-term memory of events that would otherwise not be remembered; (2) new learning during sleep phase consolidation could interfere with the consolidation. We hypothesize: (a) training during the sleep phase induces the synthesis of an active blocker of memory formation. If the blocker is removed, we predict: (b) the ongoing consolidation process will enhance the ability of training to produce long-term memory. In addition, if the ongoing consolidation is itself blocked, we predict: (c) training will no longer produce long-term memory. These predictions were tested.

## Memory is consolidated during the inactive phase

It was first important to show that *Aplysia* consolidate memory during sleep. Similar to almost all animals, *Aplysia* display a circadian rhythm of activity. They spend most of their inactive phase immobile, and relatively unresponsive to external stimuli (*Kupfermann, 1968*; *Strumwasser, 1971*). Recent work has shown that immobility during the inactive phase has characteristics of sleep (*Vorster et al., 2014*). In mammals, memory consolidation occurs in part during the inactive phase, while animals sleep (*Rasch and Born, 2013*; *Walker and Stickgold, 2006*). Since consolidation generally depends on protein synthesis (*Montarolo et al., 1986*; *Sweatt, 2010*), an effective way to examine whether consolidation occurs during sleep is to block protein synthesis at this time, and then determine whether this treatment blocks memory formation. To this end, *Aplysia* were trained with inedible food during the active period, and were treated with 10 μM of the protein synthesis inhibitor anisomycin 12 hr later, during the sleep phase. A previous study had already shown that injecting 10 μM anisomycin 10 min before training during the active phase blocked the expression of memory 24 hr later (*Levitan et al., 2010*). Thus, consolidation occurs during the hours following training. However, 10 μM anisomycin applied several hours after training did not block 24 hr memory (*Levitan et al., 2010*). Before treatment with anisomycin, we confirmed that all of the animals were immobile, and presumably were asleep. Treatment with 10 μM anisomycin, but not with the vehicle (artificial seawater - ASW) 12 hr after training blocked 24 hr memory (*Figure 1A*). Thus, after a period in which protein synthesis is not required, 24 hr memory again depends on protein synthesis, during the inactive phase, when *Aplysia* sleep. This experiment confirms that active memory consolidation occurs during sleep in *Aplysia*, and that consolidation can be interrupted by blocking protein synthesis.

## Training during the inactive phase does not produce long-term memory

Previous studies examined the efficacy of training *Aplysia* during the inactive, sleep phase (*Fernandez et al., 2003*; *Lyons et al., 2006*). Using a number of learning paradigms, including learning with inedible food (*Lyons et al., 2005*), training during the inactive sleep phase was ineffective in producing memory tested 24 hr later. We confirmed that training with inedible food during the sleep phase (3–6 hr after the change in lighting) does not produce 24 hr memory (*Figure 1B1*). In this experiment, all animals were inactive and presumably asleep before being awakened and trained. The inability to detect 24 hr memory after training during the sleep period reflects an inability to form a new memory, rather than an inability to retrieve the memory. As shown previously (*Lyons et al., 2005*), memory is readily retrieved and expressed during the inactive period, as demonstrated by the presence of memory at night 36 hr after training during the active period (*Figure 1B2*). This experiment indicates that a blocker of memory formation is active during the sleep phase. The blocker prevents experiences that would have given rise to memory during the active phase from doing so during the sleep phase (*Figure 1B1*).

## A reminder rescues memory

Ineffective experiences during consolidation can capture products synthesized by the consolidation process and become effective in producing long-term memory. Consolidation processes during sleep follows training with inedible food. We tested whether an abbreviated training that does not give rise to memory will do so when experienced during sleep phase consolidation.

In a previous study, we showed that a brief 3 min training with inedible food is too short to elicit memory formation (*Levitan et al., 2010*). However, a 3 min training is meaningful to the animals, in that it retrieves a memory. When the memory is retrieved after it has already been consolidated, the retrieval causes the memory to become labile, so that inhibition of protein synthesis can again block

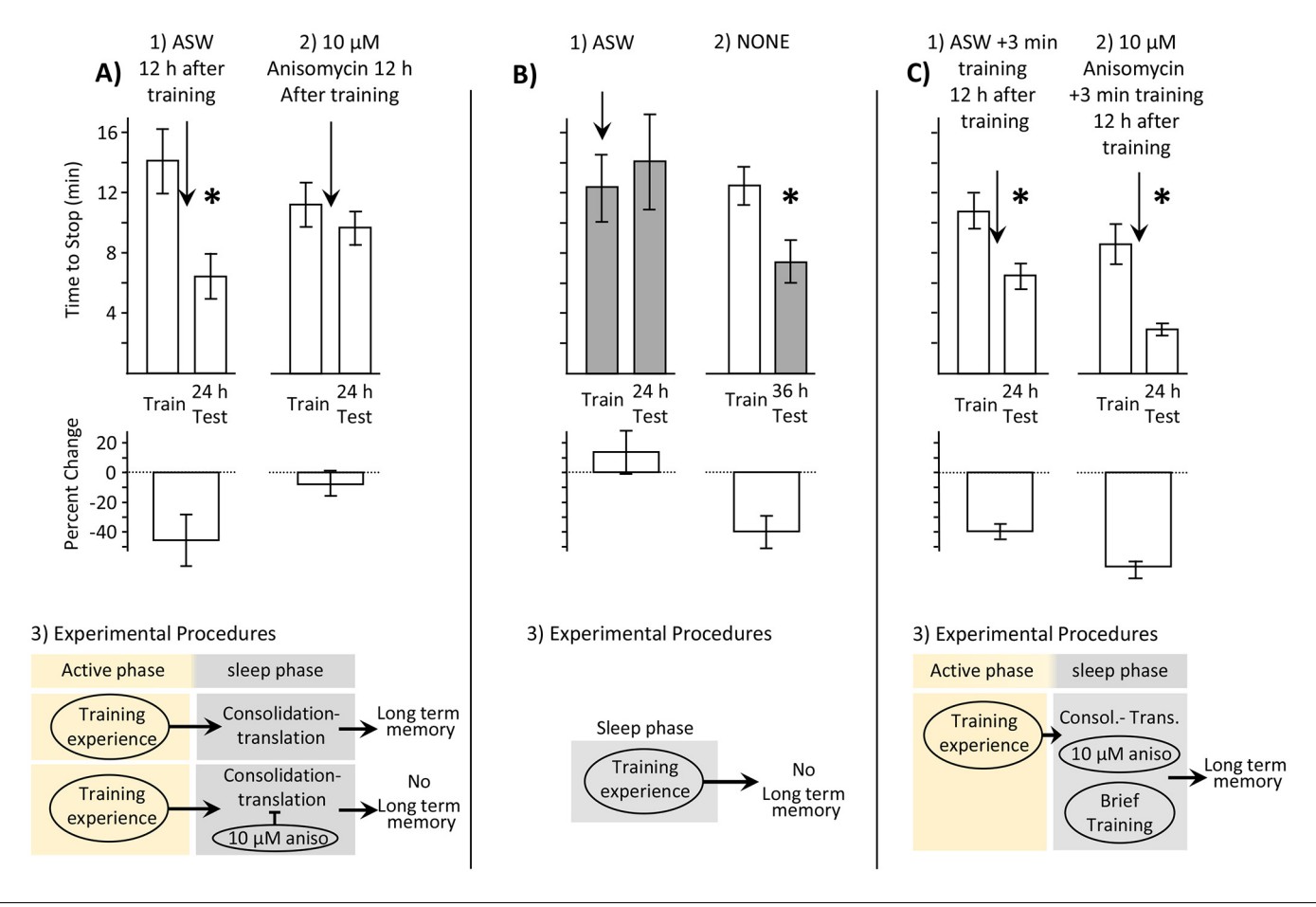

**Figure 1.** Memory is consolidated, but training is ineffective, during the sleep phase. In this and in subsequent figures, values from measurements obtained during the inactive and active phases are respectively shaded and unshaded. In this and subsequent figures, upper bars show the time to stop responding. Lower bars show the percent change in the response (-((Train-Test)/Train)*100), which is a measure of memory. In all figures standard errors are shown. Experiments in this figure were performed on the diurnally active *A. californica*. (**A**) Blocking protein synthesis during the sleep phase blocks 24 hr memory. Animals were trained during their active phase (day), and were tested 24 hr later. 12 hr after training, during the sleep phase, animals were injected with ASW (*N* = 9) or with 10 µM anisomycin (*N* = 12). (1) There is a significant decrease in the time to stop responding to the food during the 24 test (p=0.02, *t* = 2.76, *df* = 8; two tailed paired *t*-test), indicating memory. (2) There was no significant difference between time to stop responding during the original training, and the test 24 hr later (p=0.21, *t* = 1.34, *df* = 11; two tailed paired *t*-test), indicating that block of protein synthesis in the sleep phase following training blocks memory consolidation. (3) In addition to consolidation that follows training (not shown), a second phase of consolidation occurs during sleep. This phase is blocked by the translation blocker anisomycin, preventing the expression of long-term memory. (**B**) Training during the sleep phase is ineffective in causing long-term memory. (1) Training animals during the sleep phase (*N* = 5) did not lead to long-term memory, as shown by a lack of savings when animals were tested 24 hr later (p=0.3, *t* = 1.12, *df* = 4; two-tailed paired *t*-test). In this experiment, animals were treated with ASW just before the training. (2) Memory after training during the active phase is expressed during the sleep phase (*N* = 11), as shown by a significant reduction in the time to stop 36 hr after training, when animals are tested during the inactive phase (p=0.013, *t* = 3.03, *df* = 10; two-tailed paired *t*-test with Bonferroni correction). (3) A diagram showing that training during the sleep phase is ineffective in producing memory. (**C**) Effect of a brief recall during the sleep phase. (1) A 3 min training during the sleep phase (*N* = 7), which is an effective means of recalling a memory, leaves the memory intact (p=0.001, *t*(6) = 6.02, two-tailed paired *t*-test). (2) A 3 min training paired with 10 µM anisomycin (*N* = 6) rescues the memory that would have been blocked by the anisomycin alone (p=0.004, *t*(5) = 5.01, two-tailed paired *t*-test). (3) A flow diagram showing that the anisomycin does not block memory formation when followed by a brief training.

The following source data is available for figure 1:

**Source data 1.** Memory is consolidated, but training is ineffective, during the sleep phase.

memory (*Dudai, 2004*; *Nader, 2003*; *Nader et al., 2000*). The retrieval produces an additional consolidation-like state that has been termed reconsolidation (*Dudai, 2004*). Previous work showed that 24 hr after training, when a 3 min abbreviated training (reminder) is paired with 10 µM anisomycin, 48 hr memory is blocked (*Levitan et al., 2010*).

We tested whether a 3 min training during sleep phase consolidation could rescue a memory that would otherwise have been erased by treatment with 10 µM anisomycin. To examine this possibility, after a full training (training until animals stop responding to food) during the active phase, animals received a 3 min recall (training with inedible food stopped after 3 min) 12 hr later, during the sleep phase. Animals were injected with 10 µM anisomycin 10 min before the 3 min training. Controls were injected with ASW. Memory was retained after the 3 min training with ASW (*Figure 1C1*). In addition, the 3 min reminder training with 10 µM anisomycin was effective in protecting the memory (*Figure 1C2*), so that the anisomycin treatment no longer blocked the consolidation. Thus, the 3 min training rescued the memory that would otherwise have been blocked by the anisomycin treatment.

This experiment could be explained in two ways. The two explanations are not mutually contradictory: (1) the brief training blocked the effect of the anisomycin, and therefore the memory that was initiated by the training during the active phase was retained; (2) the brief training initiated a de novo memory process. Although a brief training is insufficient to produce memory during the active phase, it produces memory during the sleep phase, in the presence of anisomycin. If this hypothesis is correct, the 3 min training may be able to produce long-term memory even if it is not preceded by a full training during the active phase.

## Brief training during the sleep phase causes long-term memory

Consolidation-like activity occurs during sleep, even in the absence of an overt training during the previous active phase. Is the previous active phase training required for the brief training during the sleep phase with anisomycin to be effective in producing long-term memory? We tested the possibility that a 3 min training during the sleep phase after treatment with 10 µM anisomycin produces long-term memory, even without a previous active phase training. Naïve animals were trained for 3 min. They were injected with 10 µM anisomycin 10 min before the training. Memory was tested 24 hr later. For comparison, memory after training was compared to that before and after a non-abbreviated training session during the active period. Animals tested 24 hr after the brief training with anisomycin during the sleep phase responded like animals that had been previously trained during the active phase (*Figure 2A,B*). Thus, the combination of 3 factors which each alone is either ineffective in producing long-term memory (training during the sleep phase, training with a protein synthesis inhibitor, training stopped after 3 min), caused memory when combined.

Would the 3 min training during the sleep phase produce long-term memory even without the treatment with 10 µM anisomycin, or is the long-term memory dependent on the anisomycin treatment? Training animals for 3 min during the sleep phase after ASW treatment, rather than after 10 µM anisomycin, produced no memory (*Figure 2C*). We also tested the possibility that the brief training with anisomycin caused memory that is only expressed during the sleep phase. Animals were tested 12 hr after the brief training with 10 µM anisomycin, or with ASW. The 3 min training produced memory 12 hr later only when training was with 10 µM anisomycin (*Figure 2D*), but not with ASW (*Figure 2E*).

The findings suggest that training during the sleep phase initiates a protein synthesis dependent block of long-term memory formation. When the block is removed by the anisomycin treatment, learning is more effective during the sleep phase than during the active phase. Even an experience that is too brief to cause long-term memory during the active phase, a 3 min training, is effective during the sleep phase (*Figure 2F*). In addition, memory is expressed earlier. It was previously shown (*Botzer et al., 1998*) that after effective training during the day, long-term memory is expressed only 24 hr after training, and no memory is expressed at 12 hr after the training. However, 12 hr after a 3 min training with 10 µM anisomycin during the inactive phase, memory is expressed.

We have suggested that the 3 min training during sleep is effective in producing memory because of the ongoing consolidation during sleep, and the lack of a protein synthesis dependent blocker of memory formation. However, it is conceivable that the 10 µM anisomycin treatment per se makes the 3 min training effective, even without the consolidation process. Although the 3 min training alone during the active phase is ineffective in producing memory (*Levitan et al., 2010*), the combination of a 3 min training and treatment with 10 µM anisomycin might produce memory, even during

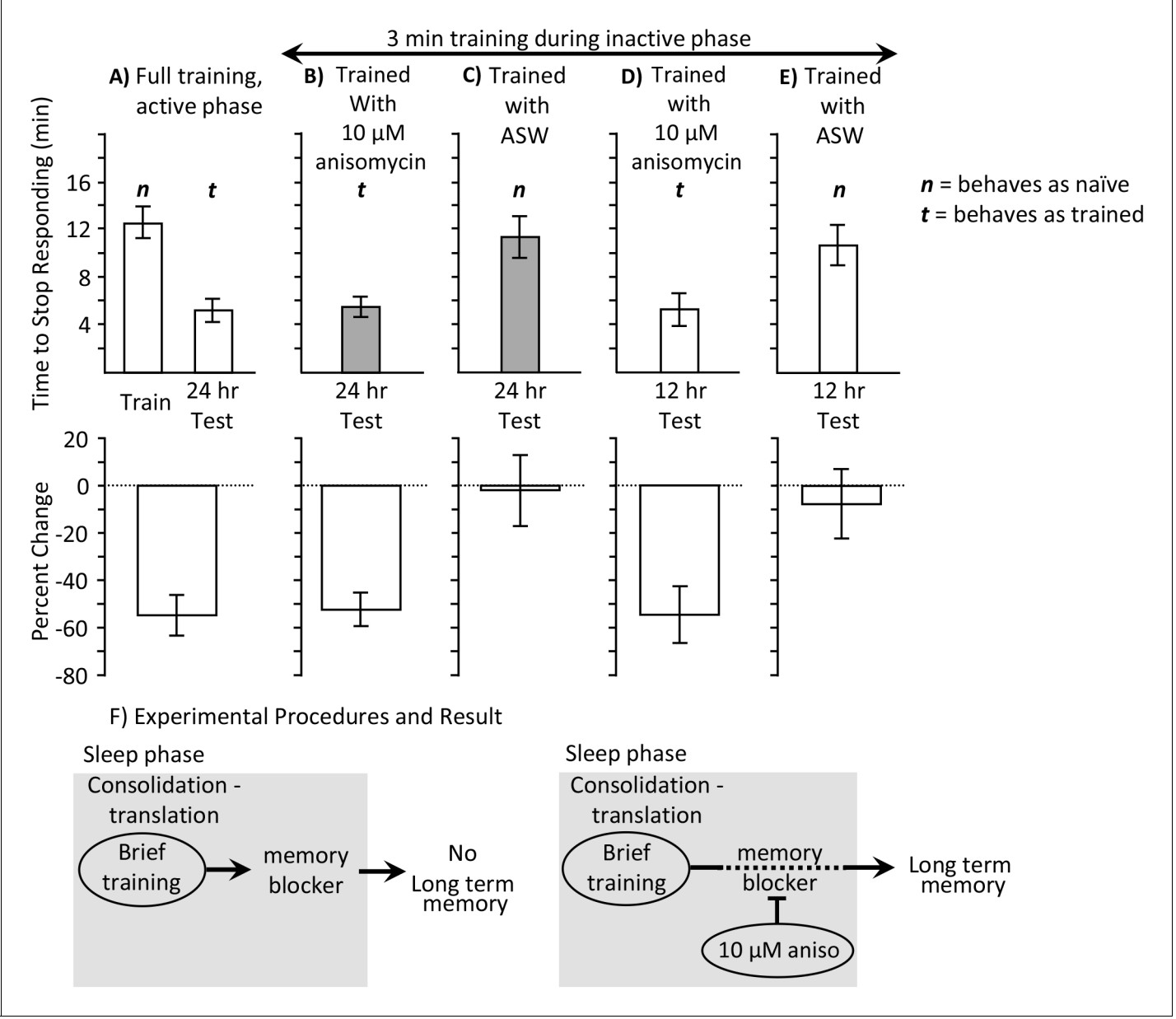

**Figure 2.** 3 min training with anisomycin during the inactive phase produces memory. Experiments were on *A. californica*. (A) Time to stop responding during training in the active phase, and during a test of memory 24 hr later (*N* = 10). These data provide comparisons for data in *B-E* on the time to stop in a naïve, previously untrained animals, and during a test of memory after successful training. (B) Animals were trained for 3 min during the inactive phase just after treatment with anisomycin, and memory was tested 24 hr later (*N* = 15). (C) As a control, animals were trained for 3 min during the inactive phase just after treatment with ASW, and memory was tested 24 hr later (*N* = 7). (D, E) To test whether memory is expressed during the active phase, animals were trained for 3 min during the inactive phase just after treatment with anisomycin (*N* = 9 D), or ASW (*N* = 11 E), and memory was tested 12 hr later. There were significant differences between the six groups tested (training and testing in part *A*, and the four tests of memory in parts B–E) (p=0.00005, *F*(5,54) = 6.95; one-way analysis of variance). A post hoc- test (Student-Newman-Keuls, $\alpha$ = 0.05) showed no significant difference between naïve animals trained during the day and either group of animals trained for 3 min with ASW (marked by an *n* for behaving as if *naïve*). Thus, a 3 min training during the inactive phase with ASW produces no memory. These 3 groups were significantly different from the other three groups (marked by a *t* for behaving as if *trained*), which were not significantly different from one another. Thus, a 3 min training during the inactive phase after anisomycin treatment produced memory 12 and 24 hr later. (F) The data are explained by the effects of 10 μM anisomycin on a process initiated by sleep phase training that blocks memory. The anisomycin prevents the action of the blocker, allowing the formation of long-term memory.

The following source data is available for figure 2:

**Source data 1.** 3 min training with anisomycin during the inactive phase produces memory.

the active phase. We tested this possibility. However, we found that the 3 min training 10 min after 10 µM anisomycin treatment during the active phase was ineffective in producing long-term memory (data not shown). Thus, this combination is effective in producing memory only during the sleep phase, presumably because of the ongoing consolidation process.

Does the ongoing consolidation in the presence of 10 µM anisomycin facilitate only a brief training, or will it also facilitate a longer training, which is effective during the active phase, but not during the sleep phase? We examined this possibility by training animals during the sleep phase until they stopped responding (full training), and testing memory 24 hr later. They were treated 10 min before training with either 10 µM anisomycin (*Figure 3A*), or with ASW (*Figure 3B*). A third group was treated with 10 µM anisomycin, but was not trained (*Figure 3C*). Only the animals that were trained after treatment with 10 µM anisomycin displayed 24 hr memory. This experiment shows that both an abbreviated and a longer training session are effective in producing memory after treatment

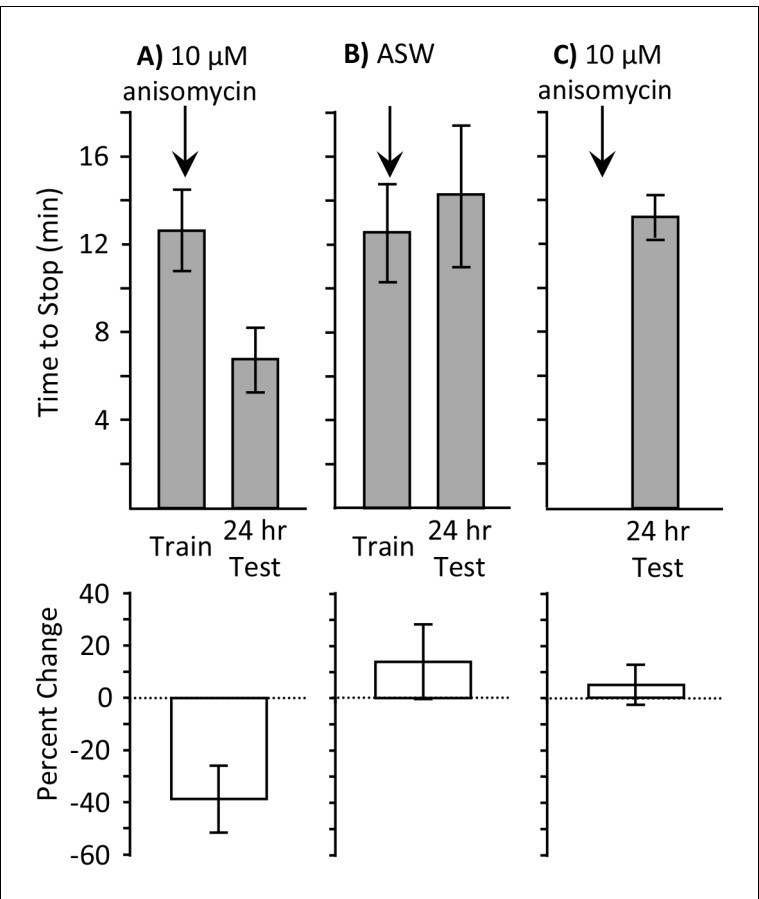

**Figure 3.** Full training during the sleep phase with anisomycin causes long-term memory. Experiments were performed on *A. californica*. (**A**) Training animals until they stop responding during the inactive phase 10 min after treatment with anisomycin (*N* = 7) leads to memory when animals are tested 24 hr later, as shown by a decrease in the time stop responding (p=0.05, *t* = 2.49, *df* = 6; two-tailed paired t-test). (**B**) Training animals until they stop responding during the inactive phase 10 min after treatment with ASW (*N* = 5) did not lead to long-term memory, as shown by a lack of savings when animals were tested 24 hr later (p=0.3, *t* = 1.12, *df* = 4; two-tailed paired t-test). (**C**) Anisomycin treatment alone during the inactive phase (*N* = 7) does not produce memory 24 hr later (p=0.74, *t* = 0.34, *df* = 16 – data after anisomycin treatment were compared to data for naïve animals trained during the day in *Figure 1B2*, which were trained along with these animals.

The following source data is available for figure 3:

**Source data 1.** Full training during the sleep phase with anisomycin causes long-term memory.

with 10 µM anisomycin. In addition it shows that anisomycin alone during the sleep phase is ineffective in producing memory without training.

## Dual function of protein synthesis during sleep

Injecting 10 µM anisomycin during the sleep phase allowed training procedures that would have been ineffective in producing long-term memory to become effective. Our interpretation is that a protein-synthesis dependent blocker of memory is inhibited by the anisomycin, and therefore is not present to suppress memory formation. However, consolidation leading to long-term memory generally requires protein synthesis. Even if it inhibited synthesis of the blocker, why was long-term memory formed when protein synthesis was blocked? We suggest that the proteins having a role in memory consolidation are either specifically expressed during sleep, or are expressed more strongly during sleep. These proteins are expressed even in the absence of an explicit training during the previous active phase, and their presence substitutes for the proteins synthesized just after training during the active phase. These proteins are synthesized throughout the sleep phase, and in our experiments they were already expressed when animals were trained later in the sleep phase. Their expression explains why training can produce long-term memory with a protein synthesis inhibitor.

This hypothesis suggests that temporally separated processes of protein synthesis explain our findings. Protein synthesis functioning in consolidation begins within the first hours of the sleep phase, and these proteins can enhance memory formation from new experiences during the sleep phase, explaining why even a brief training can produce long-term memory. However, the new experiences themselves initiate an additional process of protein synthesis that blocks the formation of new memories, explaining why training during the sleep phase is generally ineffective in causing long-term memory. Treatment with 10 µM anisomycin just before training blocks synthesis of the memory blocker, leaving intact the previously synthesized proteins associated with consolidation that enhance memory formation (see Figure 5E).

We tested this hypothesis by separately blocking the protein synthesis associated with consolidation, and the protein synthesis that blocks memory formation. We predict that blocking the protein synthesis associated with consolidation will block long-term memory formation during the sleep phase, even if the memory blocker induced by training is also not present. Since consolidation is likely to occur throughout the sleep phase, a short-acting blocker of protein synthesis that is applied early in the sleep phase may block the proteins that support consolidation, but will no longer be present at a later time, when animals are trained. In intact animals, the length of time that anisomycin effectively blocks protein synthesis depends on the concentration used. Smaller doses of anisomycin transiently block protein synthesis, whereas larger doses have longer-lasting effects (*Flood et al., 1973*). To find a transiently active dosage of anisomycin, we examined the effects of different concentrations at different times before a full training session during the active phase. We found that 10 µM anisomycin was effective in blocking memory formation 1 hr before training (*Figure 4A*), but was not effective 2 hr before training (*Figure 4B*). By contrast, a three times larger dose (30 µM anisomycin) was still effective in blocking memory formation when injected 2.5 hr before training (*Figure 4C,D*). We also tested the effect of a dosage of anisomycin that is a third of that used previously (3.3 µM in place of 10 µM anisomycin), and found that this concentration injected 10 min before training is too low to inhibit long-term memory formation (*Figure 4E*). Thus, 10 µM anisomycin is close to threshold in blocking protein synthesis, and its effect is relatively short-lasting. Injection of this concentration early in the sleep phase, 2 hr or more before training, will not affect synthesis of the proposed memory blocker, but will affect synthesis of proteins that support consolidation during sleep.

*Aplysia* were treated with 10 µM anisomycin 1 to 3 hr after the change in lighting that signaled the start of the sleep phase, but 2 hr before training began (*Figure 5A*). A second group of animals was treated with 10 µM anisomycin 30 min before the training (*Figure 5B*). Animals in both groups received a second injection of 10 µM anisomycin 10 min before a 3 min training. Our expectation is that the treatment 2 hr before training will block the synthesis of proteins associated with consolidation during sleep, without affecting the synthesis of the memory blocker. Thus, if sleep-phase proteins functioning in consolidation are required for long-term memory formation during the sleep phase, this treatment should block memory formation. By contrast, the injection 30 min before training will leave intact proteins that have already been synthesized in the early hours of the sleep phase, before the anisomycin was injected.

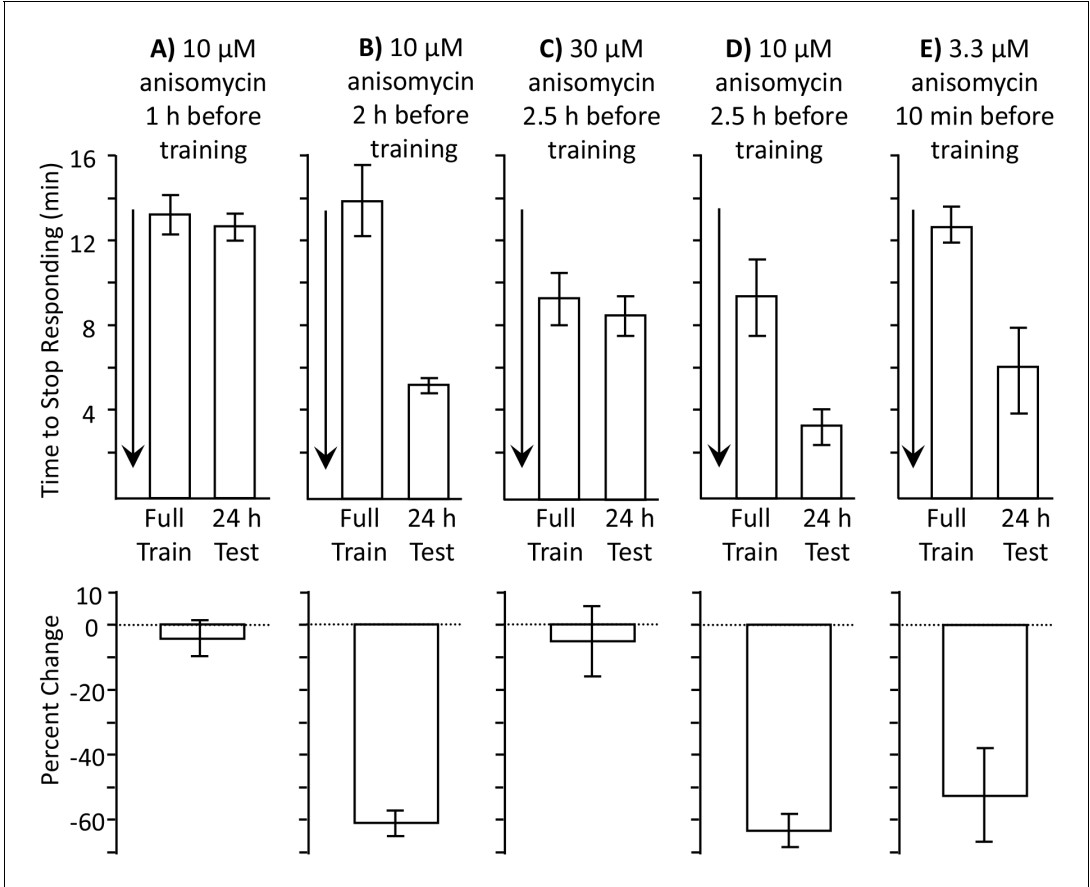

**Figure 4.** Length of protein synthesis inhibition is concentration dependent. Experiments in parts A and B were performed on *A. fasciata*; Experiments in parts C-E were performed on *A. californica*. (**A**) 10 μM anisomycin injected 1 hr before training (*N* = 5) blocks 24 hr memory, as shown by no significant change in the time to stop between the training session and the 24 hr test of memory (p=0.66, *t* = 0.48, *df* = 4; two-tailed paired t-test) (**B**) 10 μM anisomycin injected 2 hr before training (*N* = 6) does not block 24 hr memory, as shown by a significant change in the time to stop between the training session and the 24 hr test of memory (p=0.006, *t* = 4.66, *df* = 5; two-tailed paired t-test). Thus, the actions of 10 μM anisomycin are short-acting, and this concentration is no longer effective after 2 hr. (**C**) 30 μM anisomycin injected 2.5 hr before training (*N* = 6) blocks memory, as shown by no significant change in the time to stop between the training session and the 24 hr test of memory (p=0.37, *t* = 0.99, *df* = 5; two-tailed paired t-test). (**D**) In a group of animals collected and tested along with those in C, 10 μM anisomycin 2.5 hr before training (*N* = 3) did not block memory (p=0.3, *t* = 5.45, *df* = 2). Thus, 30 μM anisomycin produces a longer-lasting inhibition than does 10 μM anisomycin. For parts C and D, note that the initial training time is shorter than in parts A and B. This is because the animals were young and collected from the ocean during the beginning of the season and were also trained and tested at a different time. Baseline values often change in different batches of animals collected at different times. (**E**) 3.3 μM anisomycin injected 10 min before training (*N* = 7) is ineffective in blocking memory, as shown by a significant decrease in the time to stop between the training session and the 24 hr test of memory (p=0.01, *t* = 3.53, *df* = 6, two-tailed paired t-test).

The following source data is available for figure 4:

**Source data 1.** Length of protein synthesis inhibition is concentration dependent.

Treatment with 10 μM anisomycin 2 hr before training blocked the formation of long-term memory (*Figure 5A*), as shown by a significant increase in the response time 24 hr later, with respect to animals that were treated with anisomycin only once, 10 min before training (*Figure 5C*). By contrast, treatment with 10 μM anisomycin 30 min before training did not block the formation of long-term memory (*Figure 5B*), as shown by a significant decrease in response, with respect to the response time of naïve animals trained during the active phase (*Figure 5D*). These findings support the hypothesis that proteins synthesized during sleep substitute for the proteins that are generally synthesized as a result of long term training, and that presence of these proteins is required for long-term memory formation during the sleep phase (*Figure 5E*). This experiment also indicates that the

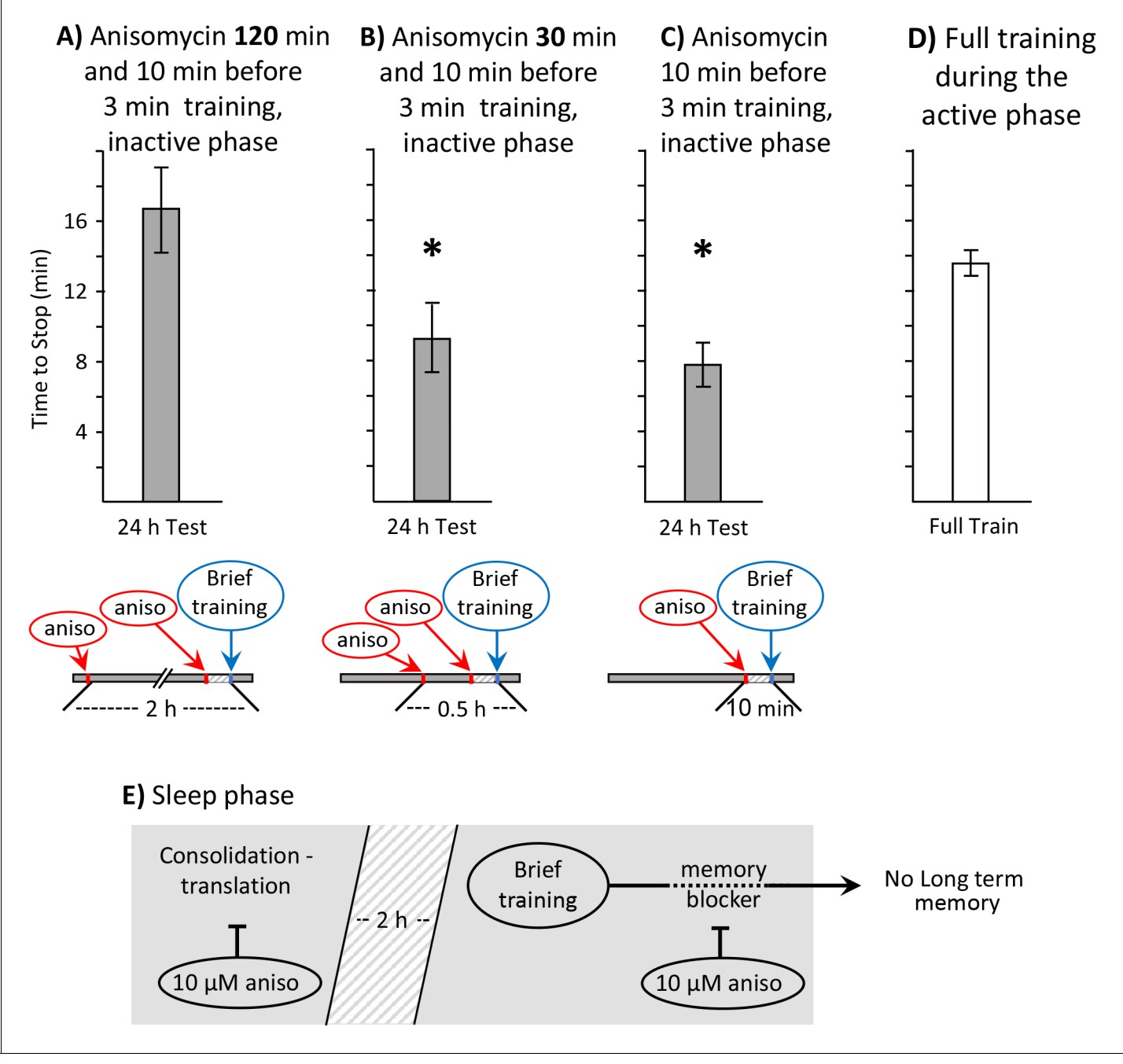

**Figure 5.** Block of sleep-time expression of proteins blocks memory formation. These experiments were on *A. fasciata*. In addition to injecting animals with 10 μM anisomycin treatment 10 min before a 3 min training, the animals were also injected with 10 μM anisomycin A) 2 hr before the training (*N* = 8), or B) 0.5 hr before the training (*N* = 8). Values 24 hr after these procedures were compared to values in C) 24 hr after a 3 min training with a single injection of 10 μM anisomycin 10 min before training (*N* = 15). A one-way analysis of variance between the three groups showed significant differences (p=0.004, $F_{(2,28)}$ = 6.90). (A) Injecting 10 μM anisomycin 2 hr before training blocked memory formation (p=0.003, *t* = 3.62, *df* = 21; two-tailed t-test with Bonferroni correction – comparison of A to C), presumably because proteins that accompany consolidation during sleep were blocked. (B) 10 μM anisomycin treatment 0.5 hr before a 3 min training plus 10 μM anisomycin did NOT block memory formation (p=0.52, *t* = 0.66, *df* = 21; two-tailed t-test – comparison of B to C), presumably because sleep-phase proteins having a role in consolidation are still present at the time of the training. (C) As shown in the previous experiments, a 3 min training during the inactive phase 10 min after 10 μM anisomycin produced 24 hr memory. (D) To provide a comparisons for data in B–D, the time to stop responding during the active phase in naïve, previously untrained animals was also measured (*N* = 6). (E) A diagram showing that translation relevant to consolidation during sleep is necessary for the expression of long-term memory after the brief training with anisomycin.

*Figure 5 continued on next page*

*Figure 5 continued*

The following source data is available for figure 5:

**Source data 1.** Block of sleep-time expression of proteins blocks memory formation.

double injection of anisomycin per se did not block memory formation, since animals injected 30 min before training, and again 10 min before training, displayed long-term memory.

These findings show that 10 μM anisomycin 2 hr before training has opposite effects during the active and inactive phases of activity. During the active phase, this treatment does not block long-term memory formation, presumably because its concentration within the animal decreases with time, and becomes too low to block the protein synthesis initiated by the training. By contrast, during the sleep phase it blocks long-term memory formation, presumably because it blocks the background expression of molecules that support memory consolidation during sleep.

## Memory formation during the sleep phase is dependent on transcription and later translation

A hallmark of long-term memory is dependence on gene transcription and translation (*Montarolo et al., 1986*). However, we have shown that memory formation at night only occurs when protein synthesis is blocked just preceding the training, although protein synthesis in the hours preceding training is necessary for long-term memory formation. Consolidation is a slow process, and is dependent on a number of waves of mRNA and protein synthesis (*Barzilai et al., 1989*). The effects of 10 μM anisomycin are relatively short-lasting (see *Figure 4*). After training, one might still expect that long-term memory formation will depend on molecular signals initiating transcription of mRNAs, and their subsequent translation, after the effects of anisomycin present during or just after training have worn off. To determine whether memory formation is dependent on transcription, we examined the effects of the reversible transcription blocker 5,6-Dichlorobenzimidazole 1-β-D-ribofuranoside (DRB) on memory formation. After treating animals with anisomycin, they were trained 10 min later for 3 min, and then treated with DRB. Memory was then tested 24 hr later.

DRB is not water soluble, and DMSO injection inhibits feeding in intact, behaving *Aplysia* (unpublished observation). We therefore used ethanol to dissolve DRB. To rule out the possibility that ethanol, or ethanol with DRB, damages animals and affects the ability to respond to food, we first examined the effects of ethanol alone and of DRB dissolved in ethanol on learning 24 hr after the injections, when their effects had worn off, and on memory 48 hr after the injection. Neither ethanol (*Figure 6A1*) nor ethanol plus DRB (*Figure 6A2*) affected the ability to train animals 24 hr after the treatment, or to obtain memory 24 hr after the training.

After a 3 min training at night with anisomycin, animals were injected with either ethanol (*Figure 6B2*), or DRB dissolved in ethanol (*Figure 6B3*). A control group also examined memory 24 hr after 10 μM anisomycin treatment at night, without subsequent training, followed by ethanol plus DRB (*Figure 6B1*). Significant memory was shown in animals treated with anisomycin alone, whereas no memory was shown in animals treated with anisomycin plus DRB. Thus, memory 24 hr after a 3 min training at night with anisomycin is dependent on transcription, and presumably also on later rounds of translation that occur after the effects of anisomycin have worn off (*Figure 6C*).

As an indicator of whether a later round of translation is required after training, we used a higher concentration of anisomycin (30 μM in place of 10 μM), and then trained animals for 3 min. The higher concentration presumably has a longer-lasting effect on protein synthesis (*Flood et al., 1973*), accounting for a longer-lasting inhibition of memory formation when *Aplysia* were trained during the active phase (see *Figure 4C*). Training with 30 μM anisomycin was ineffective in producing memory 24 hr later (*Figure 7A2*), presumably because it not only blocks protein synthesis transiently at the time of the training, but also blocks later rounds of synthesis that follow transcription (*e.g.*, *Barzilai et al., 1989*) (*Figure 6C*).

## Other effects of anisomycin are not relevant to memory formation

In addition to blocking protein synthesis, anisomycin also increases MAP-kinase activity, and thereby causes a large increase in CCAAT enhancer-binding protein (C/EBP) mRNA expression

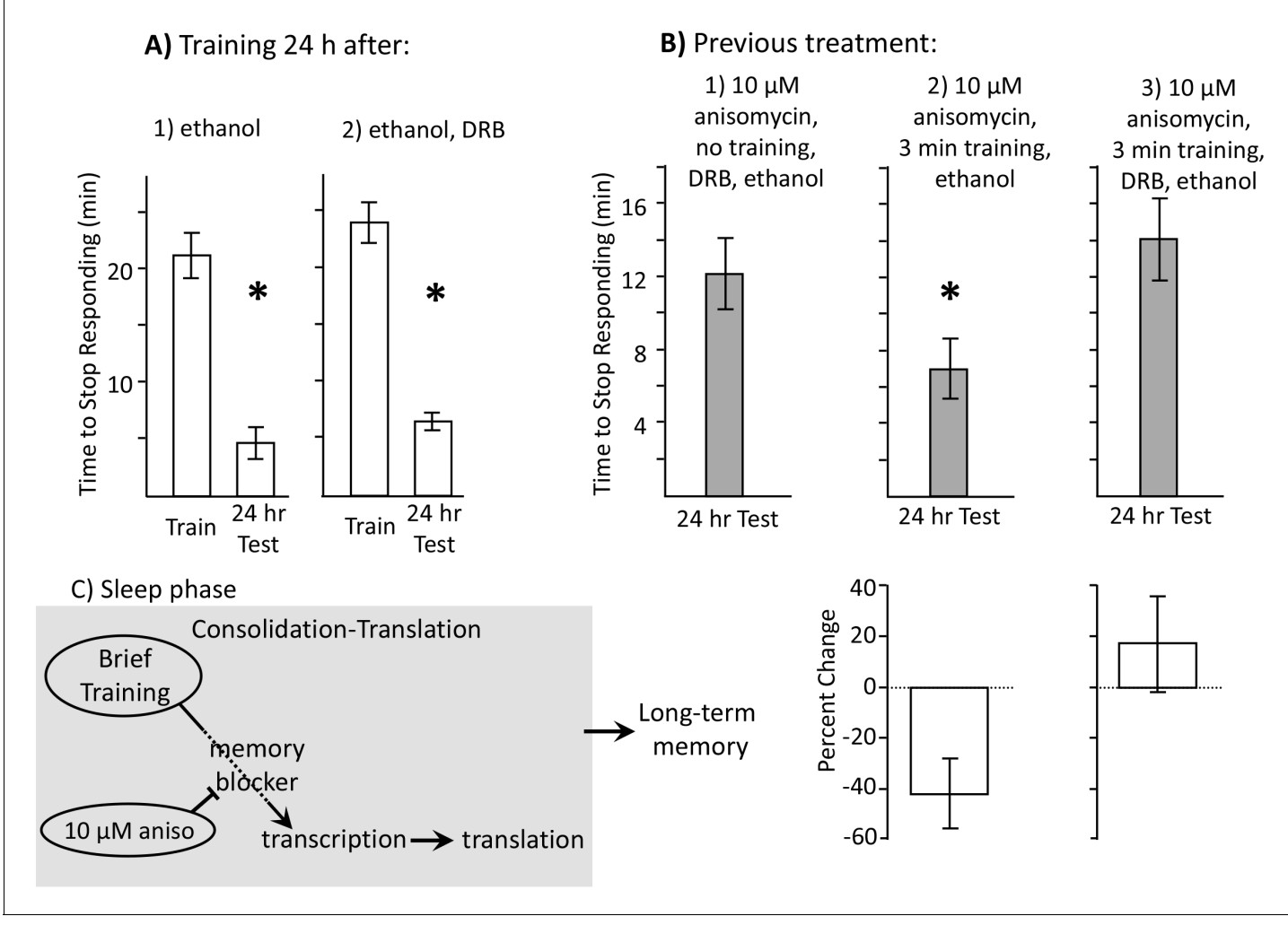

**Figure 6.** Learning during the inactive phase is transcription dependent. Experiments were on *A. californica.* (**A**) *Ethanol is benign.* Animals were treated with (1) ethanol alone (*N* = 3), or with (2) DRB dissolved in ethanol (*N* = 4), and were trained 24 hr later. Memory was then tested after 24 hr (48 hr after ethanol or DRB treatment). Neither treatment affected the ability of animals to learn or remember that food was inedible 24 hr later (For animals tested with ethanol: p=0.009, *t* = 10.22, *df* = 2; for animals tested with DRB: p=0.03, *t* = 3.57, *df* = 3). (**B**) *DRB blocks memory.* (1) *Control*: Animals were not trained. They were treated with anisomycin and then treated with DRB dissolved in ethanol, and then tested after 24 hr (*N* = 6). (2) *Vehicle*: Animals were treated with anisomycin, and then trained for 3 min. After training, the animals were injected with ethanol, with no DRB. Memory was then tested 24 hr later (*N* = 7). (3) *DRB*: Animals were treated with anisomycin, and then trained for 3 min, and then injected with DRB dissolved in ethanol. Memory was then tested 24 hr later (*N* = 8). There were significant differences between the 3 groups tested (p=0.04, *F*(2,18) = 3.73). For the 2 groups of trained animals, there was a significant difference in the time to stop 24 hr after the training (comparison of 2 and 3 – p=0.027; *t* = 2.50, *df* = 13; two-tailed *t*-test). There was no significant difference between the time to stop in animals treated with DRB, and in control animals that had not been trained (comparison of 1 and 3 p=0.49; *t* = 0.71, df = *12*; two-tailed *t*-test). Thus, the DRB blocked memory that would have been formed after training with anisomycin. (**C**) A diagram showing that after the brief training with 10 µM anisomycin, long-term memory depends on transcription and translation, The DRB blocks memory because it blocks transcription, and a 30 µM dose of anisomycin, which has a longer-lasting effect, blocks the transcription-dependent translation.

The following source data is available for figure 6:

**Source data 1.** Learning during the inactive phase is transcription dependent.

(*Alberini et al., 1994*; *Cano et al., 1994*). Increased C/EBP is an early step in the formation of memory in *Aplysia* (*Alberini et al., 1994*), and in other animals (*Alberini, 2009*). In addition, training with inedible food during the active phase caused a significant (5–10 fold) increase in C/EBP mRNA expression in the buccal ganglia (*Levitan et al., 2008*), with respect to C/EBP mRNA expression in

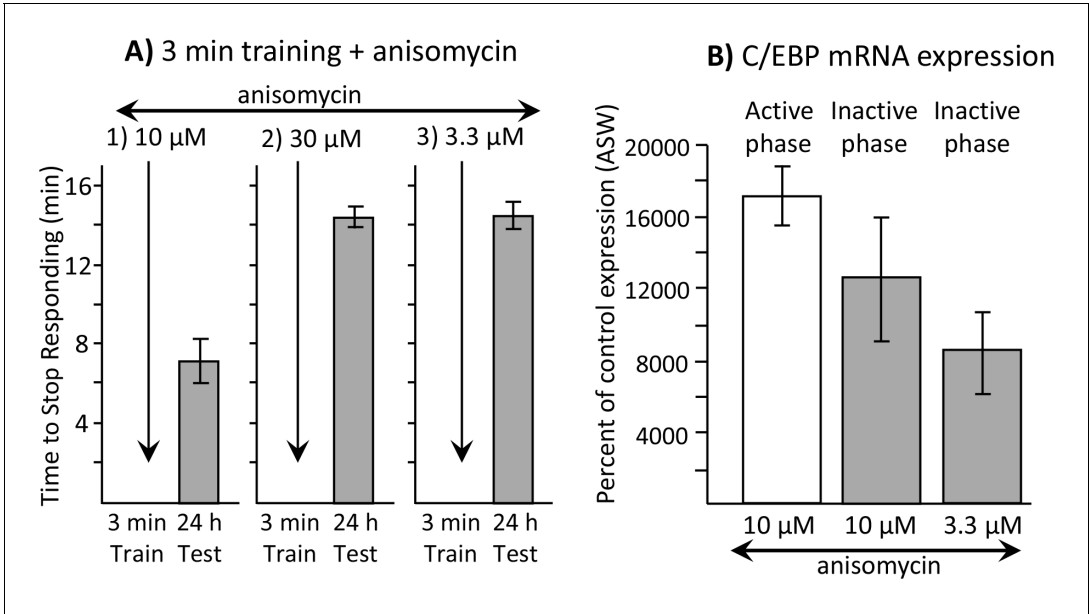

**Figure 7.** Dissociation between memory and C/EBP expression. Data are from *A. californica.* (**A**) Neither 30 µM (*N* = 6) nor 3.3 µM (*N* = 7) anisomycin injected 10 min before a brief training during the inactive phase produce 24 hr memory. The times to stop responding 24 hr after treatment with 30 µM anisomycin and 3.3 µM anisomycin are significantly longer than the time to stop 24 hr after 10 µM (*N* = 5) anisomycin (for 30 µM: p=0.005, *t* = 5.98, *df* = 9; for 3.3 µM: p=0.0002, *t* = 6.42, *df* = 10; two tailed t-tests with Bonferroni corrections). (**B**) C/EBP expression in the buccal ganglia 2 hr after injecting 10 µM anisomycin during the active (*N* = 5) and inactive (*N* = 6) phases increases, and after injecting 3.3 µM (*N* = 6) anisomycin during the inactive phase. Data are normalized to C/EBP expression in animals that were injected with ASW (not shown). Normalization was to the ASW treatment at the same time as the anisomycin treatment (*N* = 6 during the active phase, *N* = 5 during the inactive phase). There were no significant differences in C/EBP expression after ASW treatment between active and inactive phases. Expression after ASW treatment was set at 100%. All treatments with anisomycin produced large (X thousand percent) increases in C/EBP expression, although only 10 µM anisomycin during the inactive phase produces long-term memory when paired with training (For the 3 groups shown: p=0.139; *F*(2,14) = 2.27.

The following source data is available for figure 7:

**Source data 1.** Dissociation between memory and C/EBP expression.

controls whose lips had been stimulated with food, a procedure that does not lead to memory formation (*Schwarz et al., 1988*). In another *Aplysia* learning paradigm, a tyrosine phosphatase inhibitor was used to increase MAP-kinase activity, and thereby also presumably increase C/EBP mRNA expression. This treatment alone was not effective in allowing long-term memory to be formed after sleep phase training (*Lyons et al., 2006*). However, when this treatment was combined with a histone deacetylation inhibitor, the combination of both was effective in allowing long-term memory to be formed by sleep phase training (*Lyons et al., 2006*). These finding raise the possibility that anisomycin during the sleep phase permits memory formation because it increases C/EBP expression, rather than because it inhibits the synthesis of a protein that actively blocks memory formation. We tested this possibility.

Using quantitative real time polymerase chain reaction (qRT-PCR), we found that injection of 10 µM anisomycin produced a large increase in the expression of C/EBP mRNA (see *Figure 7B*). As shown above (*Figure 3C*), 10 µM anisomycin treatment alone, with no training, has no effect on subsequent memory, indicating that increased C/EBP mRNA expression per se does not produce long-term memory. However, it is possible that increased C/EBP mRNA expression underlies the ability of a brief training during the sleep phase to produce long-term memory, rather than the block of protein synthesis produced by 10 µM anisomycin. We examined this possibility.

Injection of 10 µM anisomycin 10 min before training during the active phase blocks memory, but during the sleep phase it facilitates memory. If increased C/EBP is responsible for the facilitation of long-term memory during sleep, there should be a difference in the C/EBP expression in the buccal ganglia after 10 µM anisomycin treatment during the active and sleep phases. Using qRT-PCR, we

measured C/EBP mRNA expression in the buccal ganglia 2 hr after anisomycin treatment during both the active and the sleep phases. Expression was compared to that in response to animals injected with ASW, during the active and the sleep phases respectively. During both the active and sleep phases, there were large (over 100-fold) increases in C/EBP mRNA expression in response to 10 µM anisomycin (*Figure 7B*), with no significant difference in expression between the two phases. Thus, C/EBP is increased when anisomycin blocks memory, as well as when it is required to form long-term memory.

We also tested the effects of a 3.3 µM dose of anisomycin. We found that 3.3 µM anisomycin injected 10 min before training during the active phase did not block memory formation (see *Figure 4E*). In addition, it did not facilitate memory formation during the inactive phase (*Figure 7A3*), presumably because the dose is too low to block protein synthesis sufficiently. However, 3.3 µM anisomycin still caused a large (over 80-fold) increase in C/EBP expression (*Figure 7B*). The difference between C/EBP production during the sleep phase in response to 10 µM or 3.3 µM anisomycin was not significant. Thus, there is a dissociation between the effect of anisomycin on protein synthesis, which is likely to be critical for its effects on memory, and the effect of anisomycin on C/EBP, whose increase is not systematically associated with the ability to form long-term memory.

Although increases in C/EBP caused by anisomycin treatment could not explain the ability to produce long-term memory, it is possible that increases in C/EBP are nonetheless correlated with training procedures that are effective in producing long-term memory. If this is so, training with 10 µM anisomycin at night should produce an increase in C/EBP over that produced by anisomycin alone, whereas training with ASW at night should produce no increase in C/EBP expression over that produced by ASW alone. We tested this possibility by measuring C/EBP expression 2 hr after a 3 min training with ASW or with 10 µM anisomycin. A 3 min training with ASW causes no long-term memory (*Figure 2C*), but nonetheless caused a 7-fold increase in C/EBP expression (*Figure 8A1*), which is similar to the increase in C/EBP expression caused by training during the day (*Levitan et al., 2008*). A 3 min training after treatment with 10 µM anisomycin also caused a significant increase in C/EBP expression (*Figure 8A2*). Note that the increased C/EBP expression after anisomycin treatment is 2 orders of magnitude larger than that after ASW treatment, because of the effect of the anisomycin. However, in both conditions training caused an increase in C/EBP expression. These data indicate that increased C/EBP mRNA expression is not a necessary correlate of memory formation.

The data above suggest that the increase in C/EBP expression seen after training, either during the active phase (*Levitan et al., 2008*), when training causes memory, or during the inactive phase in which training does not cause memory, is a correlate of the training procedure per se, rather than being a correlate of memory formation. Additional evidence also supported this suggestion. When *Aplysia fasciata* are trained during their active phase, but in isolation from conspecifics, no long-term memory is produced (*Schwarz et al., 1998*). We found that both training in isolation, and training in the presence of conspecifics produced significant increases in C/EBP expression (*Figure 8B*). Thus, increased C/EBP expression is a correlate of the training procedure, rather than being a correlate of memory formation.

In addition to increasing C/EBP expression, anisomycin could increase the expression of additional genes associated with memory formation, and thereby induce memory during training in the sleep phase. We examined the possibility that the expression of two additional genes associated with memory formation, cAMP- response element-binding proteins 1 and 2 (CREB1 and CREB2) (*Lee et al., 2008*), are changed by doses of anisomycin that facilitate memory formation. Since CREB2 is a repressor of memory formation (*Bartsch et al., 1995*), regulation of CREB2 might be associated with the block of memory during the sleep phase. As noted above, 3.3 µM anisomycin does not block memory formation during the active phase, or facilitate memory formation during the inactive phase, whereas 10 µM anisomycin is effective in blocking memory during the active phase, and enhancing memory formation during the inactive phase. If 10 µM anisomycin were to significantly change the expression of one of these genes, over that expressed after treatment with 3.3 µM anisomycin, this change could underlie the ability of training during the inactive phase to cause long-term memory. Unlike the effect of anisomycin on C/EBP, neither 3.3 µM nor 10 µM anisomycin during the inactive phase produced significant increases in the expression of CREB1 or of CREB2, with respect to expression after treatment with ASW, although both concentrations produced moderate increases that approached significance (*Figure 9A*, and legend – note that the bars are normalized to expression after ASW, which is 100%). There were no significant differences in expression

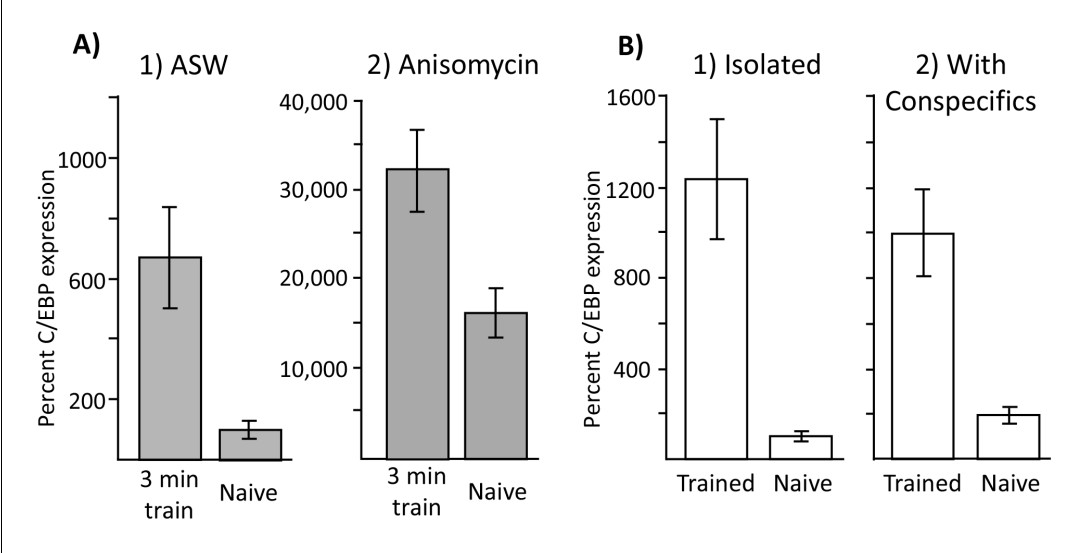

**Figure 8.** Increased C/EBP expression is a correlate of training, not of memory formation. (**A**) Effect of 3 min training during the inactive phase on C/EBP expression. Data are from the buccal ganglia of *A. californica*. (1) Training 10 min after injection with ASW (Trained: *N* = 8; Naïve: *N* = 9) produced a significant increase in C/EBP mRNA expression (p=0.004, *t* = 3.43, *df* = 15), even though this treatment did not lead to long-term memory. (2) Training after injection with anisomycin (Trained: *N* = 8; Naïve: *N* = 8) also produced a significant increase in C/EBP (p=0.008, *t* = 3.11, *df* = 14; both tests are two-tailed unpaired *t*-tests), over that caused by the injection of anisomycin in naïve controls (see *Figure 6*) Note that values for all 4 treatments are normalized to the value for naïve animals treated with ASW. As shown above, anisomycin alone caused a large increase in expression. (**B**) Effects of training in isolation on C/EBP expression. Data are from the buccal ganglia of *A. fasciata*, in which training during the active phase when animals are maintained in isolation does not produce long-term memory. Note that C/EBP expression in all four groups was normalized to expression in isolated, naïve animals, in which the mean value was set as 100%. (1) Training in *A. fasciata* maintained in isolation (Trained: *N* = 10; Naïve: *N* = 10) produced a significant increase in C/EBP expression (p=0.0009, *t* = 4.27, *df* = 18), as did (2) training in animals housed with conspecifics (Trained: *N* = 6; Naïve: *N* = 6) (p=0.004, *t* = 4.13, *df* = 10). Note that maintenance with conspecifics itself produced a small increase in C/EBP expression (p=0.05, *t* = 2.52, *df* = 14; all tests are two-tailed unpaired *t*-tests with Bonferroni correction).

The following source data is available for figure 8:

**Source data 1.** Increased C/EBP expression is a correlate of training, not of memory formation.

between treatment with 3.3 μM and 10 μM anisomycin (*Figure 9A*), effectively ruling out changed expression of either of these two genes as a result of anisomycin treatment alone from being the underlying cause of memory formation with anisomycin treatment during the inactive phase.

## Molecular correlates of sleep phase memory formation

We examined whether changes in gene expression after training are correlates of training, or of memory formation, after a 3 min training with anisomycin during the sleep phase. As shown above, C/EBP expression is significantly increased by a 3 min training at night, whether or not the training produces memory. Thus, its increase is a correlate of training, but not a correlate of memory formation. We also examined changes in the expression of CREB1 or CREB2 in the buccal ganglia 2 hr after a 3 min training during the inactive phase after treatment with either ASW or anisomycin. CREB1, but not CREB2 expression was significantly increased only after treatment with anisomycin (*Figure 9B and C*), indicating that increased CREB1 expression is a molecular correlate of long-term memory formation in our learning paradigm. Since C/EBP expression is downstream from CREB1 expression (*Alberini et al., 1994*; *Shifrin and Anderson, 1999*), and there is no increase in CREB1 expression in ineffective training, transcription factors other than CREB1, or the CREB1 protein that is already present and phosphorylated, must be driving expression of C/EBP in the buccal ganglia in ineffective training. Thus, the combination of anisomycin and training during the inactive phase has an effect on a specific mRNA whose product supports memory formation, CREB1. By contrast, CREB2 expression was not changed by training after treatment with either ASW or with anisomycin,

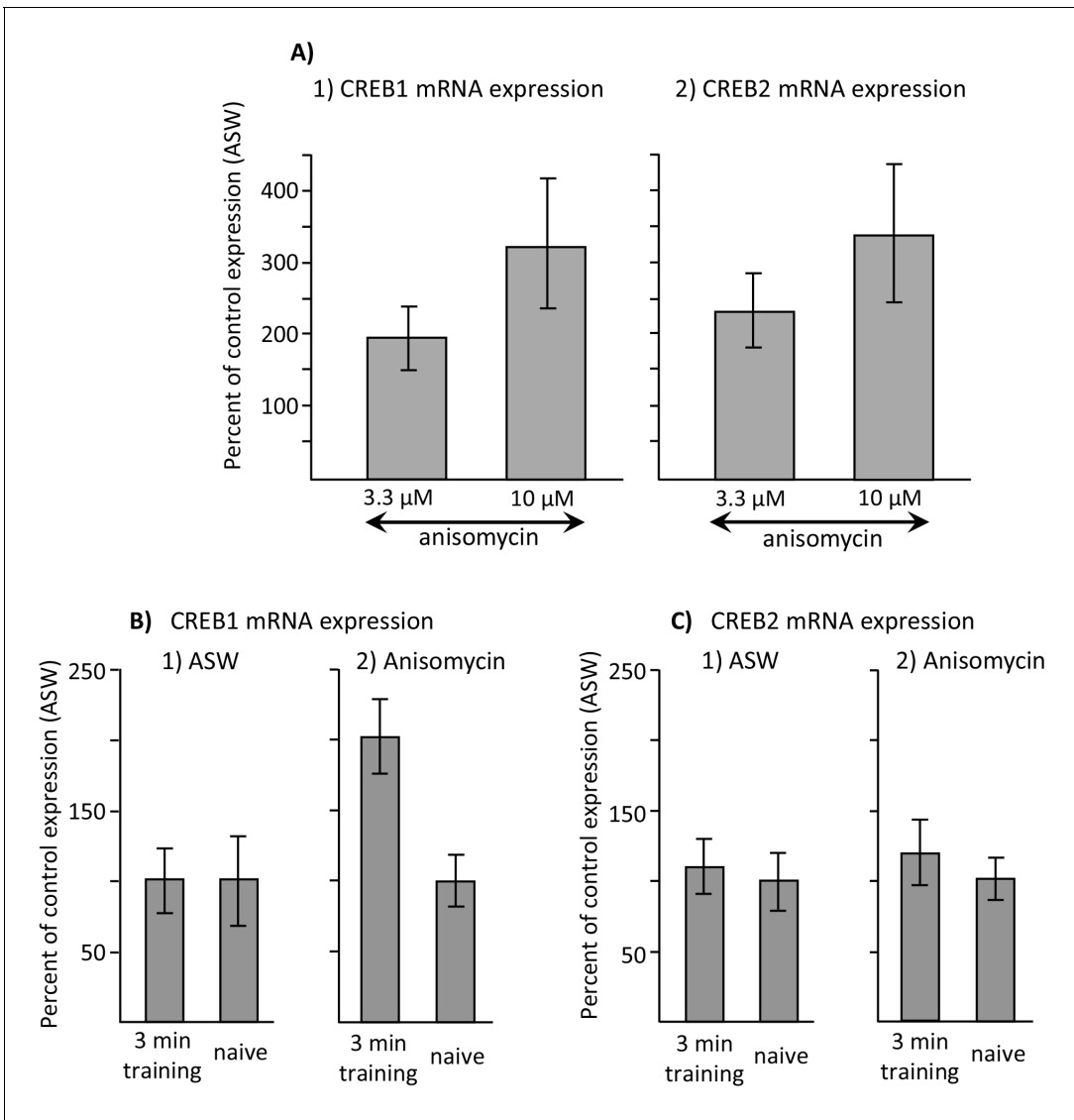

**Figure 9.** Increased CREB1 expression after training is a correlate of memory. Data are from the buccal ganglia of *A. californica*. (**A**) Increases in CREB1 or CREB2 by anisomycin treatment alone cannot account for memory formation. A preliminary analysis examined whether either CREB1 or CREB2 expression is increased by either 3.3 or 10 µM anisomycin 2 hr after treatment, with respect to expression 2 hr after injection with ASW (not shown). A one-way analyses of variance showed no significant differences in CREB1 expression between ASW (*N* = 6), 10 µM (*N* = 8) and 3.3 µM (*N* = 6) anisomycin (p=0.09, $F_{(2,17)}$ = 2.77) or of CREB2 expression between these three treatments (ASW: *N* = 6; 10 µM anisomycin: *N* = 7; and 3.3 µM: *N* = 5) anisomycin) (p=0.07, $F_{(2,15)}$ = 3.10) 2 hr after the treatments. In addition, there were no significant differences between expression after treatment with 3.3 and 10 µM anisomycin (For CREB1: p=0.29, *t* = 1.1, *df* = 12; for CREB2, p=0.42, *t* = 0.84, *df* = 10). (**B**) CREB1, but not CREB2, is a correlate of memory formation. (1) There was a significant increase (p=0.006, *t* = 3.24, *df* = 14) in CREB1 expression 2 hr after a 3 min training with 10 µM anisomycin during the inactive phase(*N* = 8), with respect to naïve animals treated with anisomycin (*N* = 8). However, there was no significant increase in expression in animals treated with ASW (p=0.97, *t* = 0.03, *df* = 15. *N* = 8 for the trained group, *N* = 9 for the naïve group). (2) CREB2 expression was unaffected by training after either treatment (For animals treated with ASW: p=0.71, *t* = 0.37, *df* = 15, N = 8 trained animals and 9 untrained animals; for animals treated with 10 µM anisomycin: p=0.50, *t* = 0.69, *df* = 12, *N* = 7 trained and 7 untrained animals). Titles and brief legends for the source data files.

The following source data is available for figure 9:

**Source data 1.** Increased CREB1 expression after training is a correlate of memory.

indicating that its expression is not altered by the training procedure, when the training does or does not produce long-term memory.

## Discussion

Experiments in which related training trials follow one another have provided major insights into the nature of memory consolidation. Indeed, consolidation was first identified by the finding that a training trial occurring during consolidation may interfere with the consolidation (*Lechner et al., 1999*). More recently, it has been found that training protocols leading to short-term memory tag synapses that are to be consolidated (*Frey and Morris, 1997*; *Martin et al., 1997*). These tags can cause a training that would only cause short-term memory to capture a consolidation process that was initiated by another training session which causes long-term memory (*Ballarini et al., 2009*; *de Carvalho Myskiw et al., 2013*; *Moncada and Viola, 2007*).

The complex interactions that may occur between an ongoing process of memory consolidation and a second, related training potentially causes problems. There are presumably good adaptive reasons why some experiences are consolidated, and cause long-term memories, whereas others are not, and cause only short-term memory, or even no memory at all. The ability of an experience to disrupt or to capture an ongoing consolidation may therefore be maladaptive.

One potential solution to the problem of maladaptive interactions between different experiences is to defer a major part of the consolidation to the sleep phase. New experiences will rarely occur during sleep, and therefore will not interfere with the consolidation. However, consolidation is unlikely to be instantly turned off when animals are awakened during the sleep phase, and non-adaptive interactions between new experiences and consolidation may then occur. In this paper, we have shown that new experiences when animals are awakened from sleep can indeed capture the ongoing consolidation process, and that an active protein synthesis dependent process blocks such capture.

When animals are trained during the active phase, protein synthesis-dependent consolidation processes occur in the hours after training, and again during the inactive phase (*Figure 1A*), when *Aplysia* sleep (e.g., *Vorster et al., 2014*). As in many other animals (*Rasch and Born, 2013*; *Walker and Stickgold, 2006*), memories are consolidated during sleep (*Figure 1A*). Blocking protein synthesis in either of these two periods blocks long-term memory formation. Even a relatively brief-acting blocker, such as 10 μM anisomycin, effectively blocks consolidation when applied either during the hour preceding training (*Figure 4A*), or during the sleep phase, 12 hr after training (*Figure 1A*).

When animals are trained during the sleep phase, no long-term memory is seen (*Figure 1B*), despite the presence of sleep-phase consolidation products, because memory formation is actively blocked. Treatment with 10 μM anisomycin removes the block, and allows training during the sleep phase to form long-term memory (*Figures 1C*, *2* and *3*). Although translation immediately following training is blocked by the anisomycin, proteins supporting molecular processes underlying memory during sleep are present, because they have been synthesized in the hours preceding the training. These substitute for the translation generally elicited by training. Blocking the expression of these proteins by treatment with 10 μM anisomycin 2 hr before training blocks long-term memory (*Figure 5*). Memory after training during the sleep phase is dependent on transcription and translation following training, since the transcription inhibitor DRB (*Figure 6B*), or a longer-lasting block of protein synthesis (30 μM anisomycin), block memory formation (*Figure 7A2*).

The ability of a training session to induce memory during the sleep phase only in the presence of anisomycin raises two questions, which will be discussed separately below: (1) why is memory formation during the sleep phase not blocked by anisomycin, as it is during the active phase? (2) Why does long-term memory after training during the sleep phase occur ONLY after treatment with anisomycin?

### Why is memory formation during the sleep phase not blocked by anisomycin?

We hypothesize that proteins which must be synthesized for memory formation as a result of training during the active phase are either already present during the sleep phase, or that other proteins present during the sleep phase substitute for proteins that are translated as a result of training. After

briefly reviewing the evidence in favor of this hypothesis, other hypotheses that could explain long-term memory formation without protein synthesis will be considered.

## Proteins required for consolidation are present during sleep

The general requirement for both transcription and translation for long-term memory formation suggests that training first induces transcription, and then proteins are translated from the transcribed mRNAs. However, aspects of memory formation also require the translation after training of proteins from pre-existing mRNAs (*e.g.*, *Ghirardi et al., 1995*; *Parsons et al., 2006*). A short-acting inhibitor of protein synthesis, such as 10 µM anisomycin, will preferentially block this synthesis, with weaker effects on later rounds of protein synthesis that arise when mRNAs induced by training are translated. We propose that proteins synthesized from pre-existing mRNAs during active-phase training do not need to be synthesized during sleep-phase training, because proteins that are already present substitute for them. This is consistent with findings in higher animals, which showed that protein synthesis, and translation initiation factors supporting protein synthesis, are increased during Slow Wave Sleep (SWS) (*Grønli et al., 2013*), the sleep phase associated with consolidation of declarative memory (*Rasch and Born, 2013*).

Why are proteins that support memory present during the *Aplysia* sleep phase? Their presence is likely to reflect their functions in the consolidation during sleep of memories that result from previous experiences. Thus, inhibiting protein synthesis during sleep disrupts memory consolidation in *Aplysia* (*Figure 1A2*), and interrupts synaptic plasticity in mammals (*Seibt et al., 2012*). However, in most of our experiments animals were not explicitly trained in the previous active phase, raising the question of why consolidation occurs in the absence of previous training. Processes related to consolidation may not occur only after an explicit training trial. Animals and humans sleep every night, even when no training occurred in the previous day, and sleep-phase consolidation products are likely to be synthesized even without overt training during the previous waking phase (*Ramm and Smith, 1990*). The consolidation may be relevant to the processing of the normal experiences of previous active phase, but even in the most boring environments in which there is little new to learn, animals sleep.

The hypothesis that proteins present during sleep which function in consolidation substitute for proteins synthesized after training during the active phase is supported by a number of findings. As noted above, inhibiting protein synthesis during the sleep phase blocks memory formation after training during the previous active phase (*Figure 1A2*). In addition, anisomycin treatment during the first hours of the sleep phase, 2 hr before training, blocks memory formation (*Figure 5A*), because the synthesis of proteins synthesized during sleep has been blocked, and they can no longer support memory formation when animals are subsequently trained. This treatment may be effective because protein synthesis was blocked before factors required for consolidation were synthesized. An alternate possibility is that during the sleep phase there is a rapid turnover of proteins supporting consolidation, and as previously synthesized proteins are removed, they are not replaced.

The synthesis of sleep-phase consolidation products predicts that training when animals are awakened from sleep should be particularly effective in producing long-term memory. This prediction is consistent with the finding that when protein synthesis is blocked just before training, even a 3 min training produces long-term memory (*Figure 2B*). During the active phase, a 3 min training is too brief to produce long-term memory (*Levitan et al., 2010*).

The finding that a training which is ineffective in producing memory becomes effective if it occurs during an ongoing process of memory consolidation has a number of precedents. In synaptic and behavioral tagging, even a weak stimulus that is insufficient to produce long-term memory captures products supporting memory formation generated as a result of a training at another site by a training protocol that does cause long-term memory (*Ballarini et al., 2009*; *de Carvalho Myskiw et al., 2013*; *Frey and Morris, 1997*; *Martin et al., 1997*; *Moncado and Viola, 2007*). As in our experiments, generation of the memory promoting products requires translation, but generation of the tags that capture the memory promoting products does not require translation. In addition, molecular cascades initiated by brief training sessions become more effective in inducing memory when they are timed to training sessions that initiate other, perhaps longer-lasting molecular cascades (*Zhang et al., 2011*). In our experiments, the translation dependent products that are required for memory formation are provided by molecular events that occur during sleep, rather than by an earlier training session.

## Increased C/EBP mRNA synthesis

A second possible mechanism explaining why anisomycin does not block memory formation during the sleep phase is that in addition to blocking protein synthesis, anisomycin also causes a large increase in the expression of C/EBP mRNA (*Figure 7B*). Previous studies (*Alberini et al., 1994*) showed that anisomycin activates MAP-kinase (in particular, stress-related p46/54$^{JNK}$ and p38$^{MAPK}$) (*Dhawan et al., 1999*), and this activation probably caused the increased C/EBP mRNA expression that we observed. Overexpression of C/EBP mRNA could produce overexpression of C/EBP protein, after the blocking effects of anisomycin have worn off, and could thereby allow a single stimulus that is insufficient to produce long-term facilitation to become effective (*Lee et al., 2001*). Thus, it is possible that the increased expression of C/EBP mRNA is sufficient to overcome the block of protein synthesis, and explains the ability of training to give rise to long-term memory. The increased C/EBP mRNA expression could also explain why even a 3 min training is effective in producing memory.

It is unlikely that the increased MAP-kinase activity driving increased C/EBP transcription is alone able to explain our results, although it could contribute to them. To determine the possible contribution of the increased C/EBP transcription to learning during the sleep phase, in addition to its role as a blocker of protein synthesis, we would need to identify another short-acting protein synthesis inhibitor without the additional effects of anisomycin. Anisomycin causes increased C/EBP synthesis both during the active phase and during the sleep phase (*Figure 7B*), but blocks memory in the former (*Figure 4A*), whereas it is necessary for memory in the latter (*Figure 2B,C* and *Figure 3A*). If increased C/EBP expression explains the ability to form memory during the sleep phase in the absence of protein synthesis, it should also promote memory formation in the absence of protein synthesis during the active phase. In addition, *Lyons et al. (2006)* tested whether a tyrosine phosphatase inhibitor which pharmacologically induces an increase in MAP-kinase activity during the sleep phase restored the ability of a training procedure causing long term facilitation of withdrawal reflexes to be effective in producing long-term memory. They found that this treatment alone was insufficient to allow training during the inactive phase to be effective. Training at night was effective in producing memory only when the tyrosine phosphatase inhibitor was paired with a pharmacological agent that inhibits histone de-acetylation, which led to increased overall RNA expression. We also showed a dissociation between increased C/EBP expression and memory formation. Thus, a dose of anisomycin (3.3 µM) that is too low to block active phase memory formation (*Figure 4E*) induced an increase in C/EBP mRNA (*Figure 7B*), but did not permit memory formation after training during the sleep phase (*Figure 7A3*). In addition, C/EBP expression was increased after training in isolated *Aplysia* that are unable to form long-term memories (*Schwarz et al., 1998*) as much as it is in animals that are trained with conspecifics, which do form long-term memory (*Figure 8B*). The increased C/EBP mRNA caused by training (*Levitan et al., 2008*), and the requirement for C/EBP for memory formation in other learning tasks (*Alberini, 2009*; *Alberini et al., 1994*) indicates that increased C/EBP mRNA expression is necessary, but is not consistently sufficient for long-term memory formation, perhaps because the downstream effects of C/EBP expression are regulated.

## Long-term memory not dependent on protein synthesis

An additional possible explanation for the lack of ability of 10 µM anisomycin to block memory formation during the sleep phase is that a form of long-term memory that is not dependent on protein synthesis is expressed at this time. In *Drosophila*, ARM (anesthesia-resistant memory) can produce memory lasting for 24 hr even after protein synthesis is blocked (*Tully et al., 1994*). Although no similar long-term memory has been identified previously in *Aplysia*, it is conceivable that such a mechanism exists, and is present during the sleep phase. However, it is not clear why such memory would not also be expressed after active-phase training with a protein synthesis inhibitor. In addition, we found that the transcription inhibitor DRB (*Figure 6B*), as well as 30 µM anisomycin (*Figure 7A2*) adjacent to training, blocked memory formation during the sleep phase, as did 10 µM anisomycin 2 hr before the training (*Figure 5A*). Thus, memory formation during the sleep phase is dependent on protein synthesis a number of hours before and after training, even if it is not dependent on protein synthesis during a short period of time immediately following the training.

## Dose of anisomycin during the sleep phase required to block memory is higher than during the active phase

An additional possibility for explaining why 10 µM anisomycin blocks long-term memory formation during the active phase, but not during the sleep phase, is that higher doses of anisomycin are required to block protein synthesis during the sleep phase than during the active phase. The higher the dose of anisomycin used, the more effective it is in blocking memory (*Flood et al., 1973*). Indeed, we found that 30 µM, but not 10 µM anisomycin prevents memory formation during the sleep phase, whereas both concentrations are effective during the active phase. However, this hypothesis is highly unlikely, since 10 µM anisomycin did effectively block memory when applied 2 hr, but not 0.5 hr before training (*Figure 5A,B*). Thus, the concentration of anisomycin required to block protein synthesis is not different during the sleep phase from that required during the active phase. However, the time at which the protein synthesis blocker is present is critical. Anisomycin applied 2 hr before training is likely to block production of proteins that participate in memory consolidation, whose presence is required at the time of the training. The higher dose of anisomycin at training is likely to be effective in blocking memory formation because it also blocks protein synthesis for a longer time (see *Figure 4C,D*), and therefore affects the translation of proteins from mRNAs that are transcribed as a result of training.

## Why does memory during sleep phase occur ONLY after treatment with anisomycin?

### Active block of new memories that interfere with consolidation

We propose that when animals are awakened from sleep, training induces the production of an active blocker of memory formation. Anisomycin treatment inhibits the production of the blocker, thereby allowing the training to be effective in producing long-term memory. The presumptive memory blocker might also be expressed during the active phase, and its expression could modulate active phase training. Thus, it is possible that a 3 min training is ineffective in producing long-term memory because such training preferentially induces expression of the blocker. Only a longer training induces the expression of additional factors that overcome the block. A protein synthesis inhibitor at this time would block the blocker, but would also block synthesis of proteins required for memory formation that are synthesized after the longer training, but that are not required during the sleep phase.

Anisomycin also produces an increase in C/EBP, and perhaps in other genes that support memory. These increases might underlie the requirement for anisomycin for memory formation, rather than the inhibitory effect of anisomycin on the synthesis of a memory blocker. However, as noted above we found that C/EBP was increased by a number of treatments not producing memory. In addition, both concentrations of anisomycin that are effective and ineffective in causing memory produced similar levels of expression of CREB1 or CREB2 (*Figure 9A*). Thus, the effects of anisomycin are likely to be due to its ability to block protein synthesis, rather than on its effects on gene expression.

Why is it adaptive for training during the sleep phase to produce a blocker of memory? Blocking memory formation during the inactive phase acts to prevent forming non-adaptive memories that result from transient experiences. There are likely to be good adaptive reasons why a 3 min training during the active phase does not produce long-term memory, but a longer training does. However, in the absence of a blocker, long-term memory is produced in response to even transient experiences during the inactive phase.

If preventing transient experiences from forming long-term memory were the only function of the memory blocker, it should block only training sessions that are generally too short to produce long-term memory, leaving intact memories formed by longer training sessions. This does not occur: the blocker also blocks memory following a longer training that produces long-term memory during the active phase (*Figure 3*). We propose that even these memories are blocked to prevent new learning from interfering with the consolidation. This is consistent with what may be the primary function of partially deferring consolidation to sleep, the prevention of new experiences from interfering with consolidation. The earliest papers (*Jenkins and Dallenbach, 1924*) showing that sleep improves memory posited that it does so passively, by preventing interference from new experiences. In recent years it has become clear that active processes also strengthen memories while animals sleep

(*Rasch and Born, 2013*; *Walker and Stickgold, 2006*). However, the presence of active processes promoting consolidation during sleep strengthens the need to limit experiences while memories are consolidated. In addition, lack of consciousness may not be sufficient to protect the consolidating memories from interference, since one can learn during sleep (*Arzi et al., 2012*), or when awaked from sleep, as in our experiments. An active, protein synthesis dependent process that blocks long-term memory formation to new experiences is an effective solution to prevent interference with consolidation. When this process is blocked by an inhibitor of protein synthesis, new experiences are able to produce long-term memory.

In theory, a protein synthesis-dependent blocking process could be initiated by the training, or could be present throughout the sleep period, with protein synthesis continuously required to produce a blocker of memory formation. However, we favor the hypothesis that the blocker is synthesized as a result of training, since this is more parsimonious than synthesis of a blocker throughout sleep. In addition, injecting the anisomycin 10 min before training was sufficient to block the blocker. If the blocker were synthesized throughout the sleep phase, it is likely that it would still be present at high levels when training began 10 min after injecting 10 μM anisomycin.

The memory formation blocker is likely to influence a number of molecular processes related to memory. Thus, *Lyons et al. (2006)* found that for another *Aplysia* learning paradigm, long-term sensitization training, block of memory formation after inactive phase training is correlated with a number of molecular changes related to memory formation.

## Possible targets of the memory blocker

Our data provide some hints on the stages of memory formation that might, or might not, be influenced by the blocker.

The blocker is unlikely to operate by preventing acquisition of learning, since behavioral parameters of training, such as the time to stop responding, and behavioral changes during training, are essentially the same during the active and inactive phases (*Lyons et al., 2005* and *Figures 1–3*). In addition, inactive phase training until animals stop responding causes short-term memory when tested 0.5 hr after training (*Lyons et al., 2005*), indicating that second messenger cascades which produce short-term memory, and that can also begin to signal long-term memory (*Kandel, 2012*), are likely to be intact. The behavioral effects of short - and long-term memory are very similar: both are food specific, both show decreased attempts to swallow, both display a reduced time to stop responding (*Botzer et al., 1998*) and both are sensitive to blocking NO (*Katzoff et al., 2002*). The similarity in behavioral effects suggests that both short and long-term memory arise from similar synaptic changes in circuit elements that are common to the expression of both long-term and short-term memory. Thus, the blocker is likely to act on mechanisms that are specific to long-term memory formation, but at sites that support both short-term and long-term memory. The blocker is also unlikely to act on ongoing processes supporting consolidation of memories laid down during the previous day, since doing so would defeat the presumed function of the blocker. Thus, if proteins supporting memory are synthesized during the inactive phase to facilitate memory consolidation, the blocker is unlikely to operate by preventing this synthesis, or by preventing the proteins supporting memory from interacting with their targets, since such a process would also interfere with the consolidation.

Some targets of the blocker are likely to be downstream from C/EBP transcription, whose mRNA expression is increased by both ineffective and effective training (*Figure 8*). In addition, treatment with 3.3 μM anisomycin during the inactive phase caused a large increase in expression of C/EBP mRNA (*Figure 7B*), but did not induce memory after training (*Figure 7A*), perhaps because 3.3 μM anisomycin does not block protein synthesis, and therefore the blocker is still effective in preventing memory formation. Effects of the blocker downstream from MAP-kinase and C/EBP might explain why others (*Lee et al., 2001*) have found that overexpression of C/EBP, or a pharmacological increase in tyrosine kinase activity leading to increased MAPK (*Purcell et al., 2003*) and probably increased C/EBP expression enhance memory formation in other learning tasks affecting *Aplysia*, whereas we have found that even a natural increase in C/EBP mRNA expression after training may have no effect on memory formation during the inactive phase (*Figure 8A*). Presumably, the blocker was not expressed in their experiments. As noted above *Lyons et al. (2006)* pharmacologically increased tyrosine kinase activity and presumably C/EBP during the inactive phase, but this

treatment alone did not restore the ability to overcome the block of long-term sensitization training. The increase in C/EBP after training in isolated animals (*Figure 8B*) that show no memory suggests that training in isolation may produce a blocker with effects similar to those caused by training during the inactive phase.

Another possible molecular target of the blocker might be the transcription of CREB1 in response to the training, since CREB1 transcription after training was indeed not elevated when animals were trained in ASW during the inactive phase (*Figure 9B*). In other learning tasks affecting *Aplysia* CREB1 expression is maintained for many hours after training, and the maintained expression is necessary for the expression of long-term memory (*Liu et al., 2011*). CREB1 transcription is upstream from C/EBP expression, but is under feedback control, so blocking processes downstream from C/EBP could block feedback control of CREB1 expression, and thereby account for the difference in CREB1 expression in the presence and absence of memory formation. However, we found that the treatment with anisomycin caused modest increases in the expression of both CREB1 and CREB2 (*Figure 9A*), independent of training, indicating that block of CREB1 transcription cannot be the only site of action of the blocker. Given the previous findings of *Lyons et al. (2006)*, it is not surprising that the blocker may act at more than one site.

## CREB1 and CREB2

Our impetus for examining transcription of CREB1 and CREB2 was the previous finding that CREB2 is a transcription repressor (*Kandel, 2012*), and decreasing the repression enhances memory formation (*Bartsch et al., 1995*). This raised the possibility that the blocker of memory formation during the inactive phase is related to a relative increase in CREB2 expression, with respect to that of CREB1. If this were true, one might expect that CREB2 would be elevated as a result of training during the inactive phase. However, we found no evidence of elevated transcription of CREB2. It is possible that measurements of CREB2 protein might show increases correlated with the memory block.

Preliminary experiments have also examined patterns of CREB1 and CREB2 in the buccal ganglia 2 hr following a full training during the active phase (Luchinsky and Susswein, unpublished). The increase in CREB1 expression was similar in magnitude to that after a 3 min training with anisomycin during the inactive phase. In addition, there was no significant change in CREB2 expression. Thus, we have no evidence that CREB2 transcription is related to expression of the memory blocker, or that the memory blocker is also expressed during active-phase training, at least for the first few hours after training. It is important to note that CREB2 transcription could still have a major role in the regulation of memory formation in our system. *Hu et al. (2015)* showed that post-synaptic increases in CREB2 and c-jun underlie a persistent increase in synaptic strength, along with presynaptic decreases. In our behavioral paradigm, a full training during the active phase produces separable long-term (24 hr) and persistent (48 hr and longer) memories (*Levitan et al., 2010*). The persistent memory after training with inedible food requires protein synthesis 6 hr after training, which is not required for 24 hr memory (*Levitan et al., 2010*). We have not determined whether the brief training with anisomycin during the inactive phase produces only 24 hr memory, or also a more persistent memory. If the training produces a persistent memory, it could require changes in CREB2 expression 6 hr or more after training, similar to the post-synaptic increases seen by *Hu et al. (2015)* for persistent synaptic plasticity. However, since they found that CREB2 expression decreases presynaptically but increases postsynaptically, examining expression in the whole buccal ganglia might not find these changes, if presynaptic and postsynaptic changes balance one another.

## Comparison with mammalian learning

The data on *Aplysia* training and memory consolidation during the sleep phase are consistent with mammalian data showing that memory consolidation occurs during sleep (*Rasch and Born, 2013*). *Aplysia* spend most of their inactive phase immobile, and relatively unresponsive to external stimuli (*Kupfermann, 1968*). Recent work has shown that immobility during the inactive phase has all of the characteristics of sleep (*Vorster et al., 2014*). Thus, when animals are roused and not permitted to rest during the inactive phase, they compensate the following day. In addition, preventing *Aplysia* from sleeping interferes with memory (*Krishnan et al., 2016*).

A number of recent experiments on mammals are consistent with the hypothesis that sleep is accompanied by an active block of learning, presumably to preserve consolidation. In humans, an

odor stimulus was used to reactivate a memory while awake, or during slow-wave sleep (SWS). The reactivation was followed by an interference task. Reactivation and interference destabilized memory while awake, but not during SWS (*Diekelmann et al., 2011*). In another study, after an odor training during the waking period, rats were exposed to an artificial pattern of stimulation of the olfactory bulb during waking or during SWS. The pattern disrupted memory when animals were awake, but during sleep induced generalization of memory, rather than disrupting it (*Barnes and Wilson, 2014*). Thus, in both studies the disruptive effects of a stimulus were actively blocked during sleep. In these experiments, an external stimulus was used to trigger an experience. However, most experiences during sleep are not initiated by outward events. These experiences (dreams) are not easily remembered, perhaps because they initiate the synthesis of a blocker, even if they are part of the consolidation process. Our current work suggests that blocking of memory formation is a protein synthesis dependent process, and its presence in *Aplysia* indicates that blocking mechanisms have deep roots in the phylogeny of all animals that sleep.

## Materials and methods

### Animals

Experiments were performed on *Aplysia californica* weighing 75–150 g that were purchased from either Marinus Scientific (Garden Grove, CA) or from South Coast Bio-Marine (San Pedro, CA), and on *Aplysia fasciata* collected along the Mediterranean coasts of Israel. The animals were stored in 600 liter tanks of aerated, filtered Mediterranean seawater maintained at 17°C. Lighting was L:D 12:12. Animals were fed 2–3 times weekly with *Ulva lactuca*, which was collected at various sites along the Mediterranean coast of Israel, or purchased from Seakura, Israel (http://www.seakura.net/), and then stored frozen.

*A. californica* are diurnally active, whereas *A. fasciata* are nocturnally active (*Lyons et al., 2005*). Training and testing during the respective active or inactive phases were opposite in the 2 *Aplysia* species. *A. fasciata* are found locally only during the summer months (*Gev et al., 1984*). When *A. fasciata* were unavailable, which was most of the year, *A. californica* were used. Experiments that were started using one *Aplysia* species were completed using that species, with no mixing of individuals of the two species in the same experiments. The species used in each experiment is noted in the relevant Figure Legend. There were no differences in training and testing procedures between the two species, and data gathered from the two species were comparable. All experiments utilizing qPCR were on *A. californica*, except for that shown in *Figure 8B*, which is from an earlier series of experiments on *A. fasciata*.

Many previous experiments have shown that statistically significant savings after training are readily obtained with samples of 6–10 individuals per treatment. For this reason, both behavioral and molecular experiments were performed using *N*s of this size. In some cases, multiple replications of the same experimental procedure were combined, producing larger *N*s.

### Timing of training

Training during the inactive phase commenced from 3 to 6 hr after the onset or offset of light. In the dark, animals were observed during training with a red LED (Kingbright LED L53SRC-E, peak wavelength = 660 nm, dominant wavelength = 640 nm) approximately 1 M from the animals. The *Aplysia* eye responds well to wavelengths from 400 to 600 nm, and the response then drops precipitously to higher wavelengths, responding poorly at the wavelengths that we used (*Waser, 1968*). Animals did not respond to the onset of the illumination. Before training during the sleep phase, animals were generally immobile, which is an external sign of sleeping (*Vorster et al., 2014*). Training and testing during the active phase was from 3 to 9 hr after light onset or offset.

### Training procedure

As in numerous previous studies examining learning that food is inedible in *Aplysia* (*Botzer et al., 1998*; *Katzoff et al., 2002, 2006*; *Levitan et al., 2012*), 24 hr before being trained animals were transferred to 10 L experimental aquaria that were maintained at room temperature (23°C). They were kept two to an aquarium, with the two animals separated by a partition allowing the flow of water. As in previous studies (*Susswein et al., 1986*), the animals were trained with inedible food,

the seaweed *Ulva* wrapped in plastic net. The food induced biting, leading to food entering the buccal cavity, where it induced attempts to swallow. Netted food cannot be swallowed, and it produces repetitive failed swallows. When the unswallowed food subsequently leaves the buccal cavity, the experimenter continues holding it touching the lips, inducing further bites, entries into the buccal cavity, and failed swallows. As training proceeds many bites fail to cause entry of food into the mouth. When food does enter the mouth, it stays within for progressively shorter periods, eliciting fewer attempted swallows. In some experiments, training proceeded until the animals stopped responding to food, which was defined as a lack of entry of food into the mouth for 3 min. In experiments in which the initial training was continued until the animals stopped responding (full training), data were included only from animals in which food was in the mouth eliciting failed attempts to swallow for at least 100 s, since previous experience (*Levitan et al., 2012*) showed that such animals almost always show long-term memory. Animals in which food was not in the mouth for 100 s during a full training were discarded. A full training session until animals stop responding to food requires 10–25 min of training. Animals that stopped responding in less than 5 min were discarded. Such a training session causes long-term memory measured after 24 hr. In other experiments, training was terminated 3 min after the first response to food. In these experiments, criterion for inclusion in the experiment was 50 s of food in the mouth.

In all experiments, testing of memory was performed using a blind procedure. After training, animals were coded, and their positions changed by a person not involved in the experiments, who kept the code, and revealed the identity of the animals to the experimenter only after the conclusion of the experiment. Blind procedures sometimes required repeating control procedures with known results, simply to have extra groups of animals, to maintain the blind procedure.

## Pharmacological agents

In many experiments, the protein synthesis inhibitor anisomycin was injected into animals at various times before training. Animals which showed stress as a result of the injection, and which inked profusely, were discarded. The specific time before training at which the anisomycin was injected was often part of the experimental design, and is always noted when the experiment is described. Animals were generally injected with a 1 cc solution of anisomycin at a concentration that caused a 10 μM concentration within the animals. The volume of the whole animal was measured by displacement in a beaker of seawater, and this volume was considered to be the volume of the solvent. This concentration blocks protein synthesis in ganglia (*Schwartz et al., 1971*). Controls were injected with one cc artificial seawater (ASW - NaCl 460 mM, KCl 10 mM, $CaCl_2$ 11 mM, $MgCl_2$ 55 mM and $NaHCO_3$ 5 mM). In some experiment, animals were injected with anisomycin so as to achieve concentrations within the animal of either 3.3 μM or 30 μM. In some experiments, the reversible transcription inhibitor 5,6-Dichlorobenzimidazole 1-$\beta$-D-ribofuranoside (DRB) was injected into animals within 2–3 min following training. DRB is not soluble in an aqueous solution, and in previous experiments in which it was used in dissected tissues it was dissolved in DMSO (*Raju et al., 1991*). We have found that injecting even one cc of DMSO into behaving animals inhibits feeding. For this reason, ethanol was used as a solvent. DRB (3 mg) was dissolved in 180 μl of absolute ethanol, and then was diluted to a volume of 3 ml with ASW, and injected into 100 g animals. Dosage was adjusted when smaller or larger animals were used.

## qRT-PCR

Quantitative real-time PCR (qRT-PCR) was used to examined whether training for 3 min with anisomycin or ASW at night increased the expression of *Aplysia* C/EBP, CREB1 and CREB2 (*Ap*C/EBP, *Ap*CREB1 and *Ap*CREB2) mRNAs. In most experiments, the expression of target mRNAs was normalized to the expression of Glyceraldehyde 3-phosphate dehydrogenase (GAPDH) mRNA, whose expression is not thought to be regulated by training, and which has been used extensively as a housekeeping control gene (*e.g.*, *Hu et al., 2015*). In a single experiment (that shown in *Figure 8B*) that was performed much earlier than the rest of the experiments, histone H4 was used as the housekeeping gene. This gene was also used in previous work as a housekeeping gene (*Levitan et al., 2008*). The value of C/EBP/GAPDH, CREB1/GAPDH or of CREB2/GAPDH obtained for each experimental or control animal was further normalized and expressed as a percentage of the mean value of the normalized gene expression in control, untrained animals run in the same

experiment, which was set at 100%. Thus, each ganglion from an untrained animal has a different value, but the mean of all these values was set at 100%. Naïve animals, and animals that were to be trained, were handled identically.

Individual ganglia were rapidly excised 120 min after training. Dissected tissues were maintained in RNA Save solution (Biological Industries Israel Beit Haemek Ltd.) at −80°C. Total RNA was extracted using EZ-RNA (Biological Industries Israel Beit Haemek Ltd.). DNA contamination was eliminated using DNA-free DNAse (Ambion). Total RNA concentration was evaluated using Thermo Scientific NanoDrop 2000c UV-Vis spectrophotometer. 200 ng of total RNA from each sample was reverse-transcribed to cDNA for qPCR analyze. Reverse transcriptase was applied using a high-capacity cDNA archive kit (RevertAid H Minus First Strand cDNA synthesis kit, Thermo Scientific). Samples were analyzed in triplicate using an Applied Biosystems StepOnePlus Real-Time PCR Systems. If one of the 3 samples deviated from the other 2 by more than 0.12 cycles, the outlier was discarded. Real-time PCR was performed using ABsolute Blue qPCR SYBR Green ROX Mix (Thermo Scientific) with the following specific primers: ApC/EBP, forward primer, 5'-GCAACTCAGCAACG-CAACAAATGC-3'; reverse primer, 5'-TTTAGCGGAGATGTGGCATGGAGT-3'. ApCREB1, forward primer, 5'-TGACAAACGCTAGTCCAACCTCAG-3'; reverse primer, 5'-CCTGACGTCATGACAACACC TTGA-3'. CREB2, forward primer, 5'-CTACGATGGAGCTGGACCTTTGG-3'; reverse primer, 5'-AGGGTTCCAACTTCAGTGTAGCG-3'. H4, forward Primer, 5'-GGTGGTGTGAAGCGTATTTCTGGT-3'; reverse primer, 5'-GGCCTTGACGTTTGAGAGCATAGA-3'. ApGAPDH, forward primer, 5'-AAGGGCATCTTGGCCTACAC-3'; reverse primer, 5'-CGGCGTACATGTGCTTGATG-3'. Analysis of mRNA levels was done using the comparative Ct method (*Livak and Schmittgen, 2001*). The *Aplysia* CREB1 gene is transcribed into 2 mRNA isomeres, CREB1$\alpha$ and CREB1$\beta$. (*Bartsch et al., 1998*). The primers that we used span both the CREB1$\alpha$ and CREB1$\beta$ sequences, and will react to both mRNAs.

Analyses of mRNA expression were performed on buccal ganglia 2 hr after training. The sole exception were the data shown in *Figure 8B*, which were examined 15 min after training.

## Acknowledgements

Supported by Israel Science Foundation Grant 1379/12. We thank Eliezer Costi for designing the night lighting system, and Hillel Chiel, Itay Hurwitz and Galit Ophir for discussions and comments on the paper.

## Additional information

### Funding

| Funder | Grant reference number | Author |
| --- | --- | --- |
| Israel Science Foundation | Grant 1379/12 | Abraham J Susswein |

The funders had no role in study design, data collection and interpretation, or the decision to submit the work for publication.

### Author contributions

RL, Conception and design, Acquisition of data, Analysis and interpretation of data, Drafting or revising the article; DL, Acquisition of data, Analysis and interpretation of data, Drafting or revising the article; AJS, Conception and design, Analysis and interpretation of data, Drafting or revising the article

### Author ORCIDs

Abraham J Susswein, http://orcid.org/0000-0001-6107-5901

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
