## [Decision Letter]

Thank you for submitting your article "New learning while consolidating memory during sleep is actively blocked by a protein synthesis dependent process" for consideration by *eLife*. Your article has been favorably evaluated by K VijayRaghavan as the Senior Editor and two reviewers, including Mani Ramaswami (Reviewer #1) - who is a member of our Board of Reviewing Editors - and David L Glanzman (Reviewer #2).

The reviewers have discussed the reviews with one another and the Reviewing Editor has drafted this decision to help you prepare a revised submission.

Summary:

Susswein and colleagues perform a series of experiments that together contribute a set of unique and interesting observations on processes that occur during sleep that are involved in regulation of new memory formation and memory consolidation. The paper examines the role of protein synthesis in new learning during an inactive, sleep-like behavioral state in the marine mollusk *Aplysia*. The authors use a learning protocol (which they developed previously) in which the animal learns that a food is inedible. Here, seaweed is wrapped in a plastic net and then presented to *Aplysia*. Initially, animals attempt to swallow the seaweed, but eventually, as they learn that the food cannot be swallowed, the number of attempted swallows declines. Full training proceeds until the animals cease swallowing altogether; this learning requires about 10-25 min in naïve animals. Full training produces long-term memory (LTM), which persists for ⩾ 24 h.

Previously, it has been shown that the consolidation of LTM in *Aplysia* depends on a protein synthesis-dependent early period, which occurs around the time of original training during the active behavioral phase-as well as a second consolidative period that occurs at night (in diurnal animals) during the inactive behavioral phase (hereafter, sleep). A significant conceptual problem for the dependence of memory consolidation on processes that occur during sleep, is that new experiences or neural activity during this period may be amplified by ongoing consolidation and thereby cause a spurious LTM that could interfere with the original memory. Here, the authors explore the hypothesis that new learning is actively suppressed during the sleep phase.

Key observations and conclusions made here:

1) That blocking protein translation with anisomycin during sleep blocks consolidation of 24-hour memory of experience received when *Aplysia* were active. Thus, during sleep, neurons show elevated expression of a panel of proteins and mRNAs required for normal memory consolidation. Additional experiments (see point 3 below) argue that this occurs in neurons and circuits independently of previous day's experience.

2) That neither full LTM training (nor abbreviated 3-minute training) performed when animals are awakened from sleep results in 24 hours memory. This argues for one of two conclusions: (a) that an inhibitor of new memory formation is expressed during sleep; or (b) that LTM training during sleep results in expression of a "blocker" of memory expression. (Additional experiments indicate that this blocker may act upstream of CREB1 transcription).

3) That a brief pulse of 10 μM anisomysin provided just before 3 minute training (or extended LTM training) of *Aplysia* awakened during sleep now results in robust LTM. Remarkably, identical pharmacological treatment following training in the active phase blocks LTM. This crucial observation suggests: (a) that almost the entire spectrum of proteins and mRNAs required for LTM expression are normally expressed during sleep (b) that 3-minute or full LTM training during sleep (but apparently not in waking animals) is associated with protein-synthesis dependent expression of a blocker of new memory formation. Brief inhibition of protein synthesis could perturb the expression of this blocker, and thereby allows LTM consolidation in training-activated circuits through LTM consolidation proteins naturally expressed in the sleep phase.

These results are novel and quite interesting. The experiments appear to have been carefully performed and the data are convincing. In general, the authors could work a little harder in communicating a complex series of experiments more clearly, although they have clearly tried (the cartoons in Figure 1 are particularly commendable). For the most part, the authors have considered alternative interpretations of their data. Below we provide general and specific comments that should be addressed in a revised manuscript.

Essential revisions:

1) An important question raised by these results is whether the consolidation of memory that results from full training during the active phase is potentially regulated by a protein synthesis-dependent memory blocker, as is the consolidation of memory during sleep.

1A) It might be difficult to test this idea using the authors' present methods, because aniso would disrupt the synthesis of both consolidative proteins induced by training, as well as any memory blocker. However, perhaps partial or full training in the active phase in the presence of a reduced concentration of aniso (e.g., 3.3 µM) could induce synthesis of the consolidative proteins, but disrupt that of the memory blocker?

1B) In any case, the authors should discuss the possibility that the induction of LTM due to training during the active phase is also under the control of inhibitory proteins that must be synthesized during training. For example, is there any evidence for the involvement of protein synthesis-dependent expression of CREB-2 in the induction of the LTM for this form of learning by full training when the animal is awake?

1C) Other alternative models may also be considered. Could protein synthesis potentially induce more efficiently in inhibitory elements of the learning circuit? Is it possible that 3 min training triggers synthesis of the memory blocker only in the context of the consolidative proteins that are synthesized during sleep?

2) The text should more clearly discuss why synthesis of the memory blocker by new experiences during sleep does not disrupt the ongoing consolidation of memories induced during the active phase. It appears that the authors hypothesise that the blocker acts upstream of CREB1 activation and that processes downstream of this are insensitive to the blocker. However, this should be clarified in part by discussing whether and why full training during the active phase, followed by 3 min training (without any aniso) during the subsequent sleep phase, results in less LTM at 24 h after the original training. Also, of interest is to discuss how full training during sleep degrades the LTM expressed 24-36 h after full training during the preceding active phase.

3) In a recent study Schacher and colleagues (Hu et al., 2015) have reported that the role of the expression of CREB-1 and CREB-2 in long-term facilitation (LTF) in *Aplysia* sensorimotor cocultures varies according to the time after training and, importantly, whether the transcription factors are expressed in the presynaptic or postsynaptic neurons. Thus, the pattern of CREB-1 and CREB-2 expression, and the relative importance of the expression of these transcription factors, differs according to the time after training, as well as according to the cellular compartment. In the present study the authors are examining the expression levels of these transcription factors in whole buccal ganglia, across many different cell types. They should discuss their CREB-1 and CREB-2 results in light of those of Hu et al.

4) The authors creditably give considerable attention to the possibility that their experiments are confounded by the secondary effects of anisomycin on MAPK activation and c/EBP induction. While these are well done and do indeed provide support for their interpretation of the observations, surely it would be easier and more conclusive to repeat a few key experiments (e.g. those shown in Figure 5) with a different translational inhibitor. While we appreciate the need to identify treatments that allow brief inhibition, necessary to see the main effect reported in this paper, we feel that these should be attempted unless there are overwhelming arguments against this.

5) The authors are commended again for considering memory consolidation processes in mammalian and *Drosophila* systems. However, they should consider developing at least one alternative model more closely (alluded to in point 1C as well). Namely that 24 hour LTM may be stored in a circuit that is different from one that encodes short-term memory, and that the hypothetical LTM blocker is expressed in cells that are different from ones involved in LTM storage. There is emerging evidence from the fly system in support of such a mechanism.

6) A key intellectual idea, which is correctly the focus of this paper, is that new LTM formation is actively blocked during sleep. I would like to see the Discussion expanded to consider the possible mechanistic basis for sleep regulation of the molecular process involved in LTM blocking. How do neuromodulators released in sleep interact with LTM consolidation processes?

[Editors' note: further revisions were requested prior to acceptance, as described below.]

Thank you for resubmitting your work entitled "New learning while consolidating memory during sleep is actively blocked by a protein synthesis dependent process" for further consideration at *eLife*. Your revised article has been favorably evaluated by K VijayRaghavan as the Senior editor, Mani Ramaswami, Reviewing editor, and one reviewer.

The manuscript has been improved but there are remaining issues that need to be addressed before acceptance, as outlined below:

1) The issue of whether the 3 min training induces a memory blocker is an important one. The authors argue against the need to another experiment. This argument is accepted. However, the authors should engage with the intellectual possibility and its ramifications in their Discussion.

2) The reviewers note that the authors are correct that Schwartz et al. (1971) reported that cyclohexamide, at a concentration of 60 µg/ml did not inhibit protein synthesis in intact *Aplysia*. However, later reports found that cyclohexamide, as well as puromycin, were effective in inhibiting protein synthesis in the isolated eye of *Aplysia* (Rothman and Strumwasser, 1976; Raju et al., 1990). Moreover, others have used cyclohexamide successfully to block long-term memory in the intact terrestrial snail, Limax (Matsuo et al., 2002) and intact cuttlefish, a cephalopod mollusk (Agin et al., 2003). So, the lack of availability of an alternate protein synthesis inhibitor does not seem an adequate justification for not doing the experiment. The authors are requested to reconsider their stance. The experiment is optional, but if it is not done, then the Discussion should acknowledge the minor possibility of a secondary unknown effect of the one protein-synthesis inhibitor used here.

---

## [Author Response]

[…] Our response to the review will not be in the order of the points made by the review, since the response to the first point is dependent on the response made to last point.

*[…] Essential revisions:*

*6) A key intellectual idea, which is correctly the focus of this paper, is that new LTM formation is actively blocked during sleep. I would like to see the Discussion expanded to consider the possible mechanistic basis for sleep regulation of the molecular process involved in LTM blocking. How do neuromodulators released in sleep interact with LTM consolidation processes?*

Our original submission did not discuss the possible mechanisms in detail, since we felt that such a discussion was too speculative, given the lack of hard data on the mechanism. The request of the reviewer has forced us to think through what we can nonetheless conclude about mechanism from the data already present. It is surprisingly more than we thought. The revised Discussion has a whole new section devoted to the possible mechanisms by which memory formation is blocked (subsection “Possible targets of the memory blocker”):

Briefly, the first paragraph in this section argues that the blocker cannot act by preventing acquisition of learning, or by blocking second messenger cascades that initiate both short- and long-term memory, since behavior during training and short-term memory are intact. The second paragraph argues that some targets of the blocker are likely to be downstream from C/EBP transcription. This is consistent with the demonstrated increases in C/EBP mRNA expression caused by training even when the blocker is active, and by the lack of effect of 3.3 µM anisomycin on memory formation, in spite of its large effect on C/EBP mRNA synthesis. A blocker downstream from C/EBP transcription would explain why previous reports succeeded in amplifying the ability to form long-term memory by procedures that increase MAPK and C/EBP, whereas we do not. The third paragraph deals with the possibility that increased transcription of CREB1 may be a site of regulation.

*1) An important question raised by these results is whether the consolidation of memory that results from full training during the active phase is potentially regulated by a protein synthesis-dependent memory blocker, as is the consolidation of memory during sleep.*

*1A) It might be difficult to test this idea using the authors' present methods, because aniso would disrupt the synthesis of both consolidative proteins induced by training, as well as any memory blocker. However, perhaps partial or full training in the active phase in the presence of a reduced concentration of aniso (e.g., 3.3 µM) could induce synthesis of the consolidative proteins, but disrupt that of the memory blocker?*

The proposed experiment is to treat with 3.3 µM anisomycin, and then train for 3 min during the active phase, to determine whether the treatment with the low dose of anisomycin enhances the ability to produce long-term memory. This is an interesting experiment, and we will definitely do it. However, if we get memory under these conditions, the interpretation is not at all straightforward. One possibility is that 3.3 µM anisomycin blocks a memory blocker that is also expressed during the active phase, thereby producing memory with even a brief training. This is the suggestion of the reviewer. However, another possibility is that 3.3 µM anisomycin causes a large increase in MAPK and C/EBP (as shown in Figure 8), thereby permitting even a brief training to produce long-term memory. This does not occur in the inactive phase, since the memory blocker acts downstream from MAPK and C/EBP. The memory blocker is absent during the day, and therefore allows the increased C/EBP to enhance memory formation.

The two possible interpretations of this experiment (if 3.3 µM anisomycin indeed enhances memory formation) can be separated by using phosphatase inhibitor, bpV, which enhances MAPK and C/EBP. We predict that treatment with this substance during the active phase may allow a 3 min training to produce long-term memory, but the same treatment during the inactive phase may be ineffective (as it was when applied alone for causing long-term sensitization during the inactive phase – see Lyons et al., 2006). These are a potentially important experiment since they will shed light on a target at which the memory blocker works. However, since the experiment proposed by the reviewers cannot alone resolve the question that they raise, we respectfully say that these experiments are beyond the scope of the present paper, and will be important for a follow-up that directly examines the mechanism of the blocker. The current paper is already quite long, and quite complex, and adding experiments that aim at showing that the blocker acts at least in part downstream from C/EBP will make the paper longer and more complex.

As noted above, in the revised manuscript, we have addressed ourselves to the possibility that the blocker acts downstream from C/EBP, without proving this point. This paragraph reads: "Some targets of the blocker are likely to be downstream from C/EBP transcription, whose mRNA expression is increased by both ineffective and effective training (Figure 8). […] The increase in C/EBP after training in isolated animals (Figure 8) that show no memory suggests that training in isolation may produce a blocker with effects similar to those caused by training during the inactive phase."

*1B) In any case, the authors should discuss the possibility that the induction of LTM due to training during the active phase is also under the control of inhibitory proteins that must be synthesized during training. For example, is there any evidence for the involvement of protein synthesis-dependent expression of CREB-2 in the induction of the LTM for this form of learning by full training when the animal is awake?*

We have added a section to the Discussion on CREB1 and CREB2. This section explicitly deals with the hypothesis that CREB2 could be the memory blocker, or could have a role in the block, given the previous evidence that it is a repressor of processes required for long-term memory formation, although it may not only be a repressor. The following paragraph in the Discussion directly addresses the point raised by the reviewer:

"Preliminary experiments have also examined patterns of CREB1 and CREB2 in the buccal ganglia 2 h following a full training during the active phase (Luchinsky and Susswein, unpublished). The increase in CREB1 expression was similar in magnitude to that after a 3 min training with anisomycin during the inactive phase. In addition, there was no significant change in CREB2 expression. Thus, we have no evidence that CREB2 transcription is related to expression of the memory blocker, or that the memory blocker is also expressed during active-phase training, at least for the first few hours after training."

*1C) Other alternative models may also be considered. Could protein synthesis potentially induce more efficiently in inhibitory elements of the learning circuit? Is it possible that 3 min training triggers synthesis of the memory blocker only in the context of the consolidative proteins that are synthesized during sleep?*

*2) The text should more clearly discuss why synthesis of the memory blocker by new experiences during sleep does not disrupt the ongoing consolidation of memories induced during the active phase.*

We have addressed ourselves to this point in the revised manuscript, which states: "The blocker is also unlikely to act on ongoing processes supporting consolidation of memories laid down during the previous day, since doing so would defeat the presumed function of the blocker. Thus, if proteins supporting memory are synthesized during the inactive phase to facilitate memory consolidation, the blocker is unlikely to operate by preventing this synthesis, or by preventing the proteins supporting memory from interacting with their targets, since such a process would also interfere with the consolidation."

*It appears that the authors hypothesise that the blocker acts upstream of CREB1 activation and that processes downstream of this are insensitive to the blocker.*

We do not claim this. Although we have some evidence that the blocker acts upstream to CREB1 transcription, the revised Discussion also develops the point that the blocker may act downstream from C/EBP mRNA – see above, reply to reviewer comments 1, 1A.

*However, this should be clarified in part by discussing whether and why full training during the active phase, followed by 3 min training (without any aniso) during the subsequent sleep phase, results in less LTM at 24 h after the original training.*

This comment refers to Figure 1, which show the effects on 24 h memory of a 3 min training 12 h after active phase training, with and without anisomycin. Although both show memory when tested 24 h after the training, the reviewer astutely picked up that there is a significant difference between the animals treated with anisomycin and the control. In writing the paper, we debated among ourselves whether or not to make this comparison, and if so, what it meant. Interestingly, our preliminary interpretation was that the anisomycin treatment produced extra large savings, rather than the conclusion raised by the reviewer, that 3 min alone reduced the savings. We decided *not* to draw attention to this point, since there is a fair amount of variability from replication to replication in the percent savings, and the values both with and without anisomycin are well within the range seen in other experiments. For example, the savings without anisomycin are comparable to those shown in Figure 1 for active phase learning and memory, and savings with anisomycin are comparable to those shown in in Figure 2 for active phase learning and memory. In general, over the years we have made few or no inferences about possible differences in the relative strength of savings after training, because of this variability. It would take a very large N to make a convincing argument on whether the savings when treated with anisomycin are meaningfully different from those without anisomycin. We do not think that the significant difference in Figure 1 is meaningful. In addition, if the 3 min training during the sleep phase weakened memory consolidation, the proper comparison would be to animals that have not received the 3 min training, rather than to animals that received the 3 min training with anisomycin.

*Also, of interest is to discuss how full training during sleep degrades the LTM expressed 24-36 h after full training during the preceding active phase.*

We do not have any data on how a full training during sleep affects a previous active phase training. However, the reviewer's question presupposes that a 3 min training during sleep attenuates memory produced during the previous active phase. As noted above, we do not think that such a phenomenon is present.

*3) In a recent study Schacher and colleagues (Hu et al., 2015) have reported that the role of the expression of CREB-1 and CREB-2 in long-term facilitation (LTF) in Aplysia sensorimotor cocultures varies according to the time after training and, importantly, whether the transcription factors are expressed in the presynaptic or postsynaptic neurons. Thus, the pattern of CREB-1 and CREB-2 expression, and the relative importance of the expression of these transcription factors, differs according to the time after training, as well as according to the cellular compartment. In the present study the authors are examining the expression levels of these transcription factors in whole buccal ganglia, across many different cell types. They should discuss their CREB-1 and CREB-2 results in light of those of Hu et al.*

As noted above, we have added a section to the Discussion on CREB1 and CREB2. The second paragraph of this section has an extended discussion on the work of Hu et al., and its relevance to our study. The relevant sentences read: "Thus, we have no evidence that CREB2 transcription is related to expression of the memory blocker, or that the memory blocker is also expressed during active-phase training, at least for the first few hours after training. […] However, since they found that CREB2 expression decreases presynaptically but increases postsynaptically, examining expression in the whole buccal ganglia might not find these changes, if presynaptic and postsynaptic changes balance one another."

Because we have added this section on the possible function of CREB2 in persistent memory, in the Introduction we have added a number of words that mention that the learning paradigm being studied displays separable long-term and persistent memory processes, as a set-up to this section.

*4) The authors creditably give considerable attention to the possibility that their experiments are confounded by the secondary effects of anisomycin on MAPK activation and c/EBP induction. While these are well done and do indeed provide support for their interpretation of the observations, surely it would be easier and more conclusive to repeat a few key experiments (e.g. those shown in Figure 5) with a different translational inhibitor. While we appreciate the need to identify treatments that allow brief inhibition, necessary to see the main effect reported in this paper, we feel that these should be attempted unless there are overwhelming arguments against this.*

There are indeed overwhelming arguments against this. First, Schwartz et al., 1971 tested the efficacy of a number of protein synthesis inhibitors in blocking the incorporation of labelled leucine into proteins. He "[…] found that streptomycin (0.1 mg/ml), erythromycin (0.8 mg/ml), and chloramphenicol (0.36 mg/ml), antibiotics which act specifically on the procaryotic ribosome, were without effect. Cycloheximide (60 µg/ml), emetine (2 µM), and diphtheria toxin (8 Lf/ml), which specifically inhibit protein synthesis in eucaryotes, were also ineffective, as was puromycin (80 µg/ml), even when the connective tissue sheath of the ganglion was cut open with the intention of permitting them better access to the nerve cells. Emetine at a concentration of 200 µM inhibited by only 33%. The lack of effect of diphtheria toxin, puromycin, and emetine might result from a barrier to permeation of these agents into nerve cells, and it is possible that cycloheximide is unstable in the slightly alkaline seawater. Wilson (47) has reported some inhibition of leucine incorporation into R2 with rather high concentrations of cycloheximide (0.2 mg/ml) and puromycin (1 mg/ml)."

Subsequent to this study, emetine at a higher dosage has been used to block memory consolidation in *Aplysia,* as has been rapamycin. The effects of emetine are not reversible (e.g., Grollman, A.P. Inhibitors of protein biosynthesis V. Effects of emetine on protein and nucleic acid biosynthesis in HeLa cells. J. Biol. Chem., 1968, 243, 4089-4094.), making emetine a poor choice for our experiments. Rapamycin is not soluble in seawater, and it must therefore be dissolved in DMSO or ethanol before being added to seawater saline. This is a perfectly good procedure, when working with an isolated ganglion, or in culture, but not in the intact animal. In our experience, injecting behaving animals with DMSO turns off feeding. For the experiment reported with the transcription inhibitor DRB, we overcame this problem by dissolving the DRB in ethanol. These were by far the most difficult experiments reported in the paper, since we are working with a fairly large dose of ethanol. We did these experiments with DRB, because we had little choice – we did not want to use actinomycin because its effects are not reversible, whereas the effects of DRB are reversible. It would be particularly interesting to test the effects of rapamycin, since it is not a general purpose translation inhibitor, but rather only blocks proteins that are downstream from TOR. Is the blocker downstream from TOR? Perhaps not. In any case, this is not a trivially easy experiment to perform. In our estimate, it would take us several months to do this experiment properly.

Schwartz et al., 1971 did report that 2 additional protein synthesis inhibitors were effective: pactamycin and sparsomycin. To my knowledge, these have not been used in *Aplysia* in the 45 years that have elapsed since this paper was published. In addition, there are reports that both pactamycin and sparsomycin effects are irreversible (e.g., Lima MF, Kierszenbaum F. Biochemical requirements for intracellular invasion by *Trypanosoma cruzi*: protein synthesis. J Protozool. 1982 Nov;29(4):566-70; Coutsogeorgopoulos C, Miller JT, Hann DM. Inhibitors of protein synthesis V. Irreversible interaction of antibiotics with an initiation complex. Nucleic Acids Res. 1975 Jul;2(7):1053-72.).

*5) The authors are commended again for considering memory consolidation processes in mammalian and Drosophila systems. However, they should consider developing at least one alternative model more closely (alluded to in point 1C as well). Namely that 24 hour LTM may be stored in a circuit that is different from one that encodes short-term memory, and that the hypothetical LTM blocker is expressed in cells that are different from ones involved in LTM storage. There is emerging evidence from the fly system in support of such a mechanism.*

In our system, we think that it is unlikely that STM and LTM arise and are stored in different circuits. This is because the behavioral expression of short and long term memories are very similar. In the revised Discussion, we have addressed this point: "The blocker is unlikely to operate by preventing acquisition of learning, since behavioral parameters of training, such as the time to stop responding, and behavioral changes during training, are essentially the same during the active and inactive phases (Lyons et al., 2005, and Figure 1–Figure 3). […] Thus, the blocker is likely to act on mechanisms that are specific to long-term memory formation, but at sites that support both short-term and long-term memory."

[Editors' note: further revisions were requested prior to acceptance, as described below.]

*[…] The manuscript has been improved but there are remaining issues that need to be addressed before acceptance, as outlined below:*

*1) The issue of whether the 3 min training induces a memory blocker is an important one. The authors argue against the need to another experiment. This argument is accepted. However, the authors should engage with the intellectual possibility and its ramifications in their Discussion.*

We have added the following to the Discussion:

"The presumptive memory blocker might also be expressed during the active phase, and its expression could modulate active phase training. […] A protein synthesis inhibitor at this time would block the blocker, but would also block synthesis of proteins required for memory formation that are synthesized after the longer training, but that are not required during the sleep phase."

*2) The reviewers note that the authors are correct that Schwartz et al. (1971) reported that cyclohexamide, at a concentration of 60 µg/ml did not inhibit protein synthesis in intact Aplysia. However, later reports found that cyclohexamide, as well as puromycin, were effective in inhibiting protein synthesis in the isolated eye of Aplysia (Rothman and Strumwasser, 1976; Raju et al., 1990). Moreover, others have used cyclohexamide successfully to block long-term memory in the intact terrestrial snail, Limax (Matsuo et al., 2002) and intact cuttlefish, a cephalopod mollusk (Agin et al., 2003). So, the lack of availability of an alternate protein synthesis inhibitor does not seem an adequate justification for not doing the experiment. The authors are requested to reconsider their stance. The experiment is optional, but if it is not done, then the Discussion should acknowledge the minor possibility of a secondary unknown effect of the one protein-synthesis inhibitor used here.*

We have been able to do the experiment using cycloheximide. We based ourselves on findings in: Yeung and Eskin, Responses of the Circadian System in the *Aplysia* Eye to Inhibitors of Protein Synthesis. Journal of Biological Rhythms 3: 225-236, 1988. In this paper, the authors showed that 2 mM, 5 mM and 10 mM concentrations of cycloheximide produce inhibition of amino acid incorporation, which decreases to baseline levels within 3 hours. We tested the effects of 5 mM cycloheximide in 2 intact animals. Immediately after they were injected, the animals lost contact with the substrate, floating in the seawater, with open parapodia. They remained in this position for as long as they were observed, over 6 hours. They were dead the next day. So this treatment is not appropriate to block protein synthesis in intact *Aplysia*.

We have added the sentence in bold to the Discussion:

"It is unlikely that the increased MAP-kinase activity driving increased C/EBP transcription is alone able to explain our results, although it could contribute to them. To determine the possible contribution of the increased C/EBP transcription to learning during the sleep phase, in addition to its role as a blocker of protein synthesis, we would need to identify another short-acting protein synthesis inhibitor without the additional effects of anisomycin."